# Bayesian Optimization with Preference Exploration using a Monotonic Neural Network Ensemble

**Hanyang Wang**
University of Warwick
Coventry, CV4 7AL, United Kingdom
Hanyang.Wang@warwick.ac.uk

**Juergen Branke**
University of Warwick
Coventry, CV4 7AL, United Kingdom
Juergen.Branke@wbs.ac.uk

**Matthias Poloczek**[*]
Amazon
San Francisco, CA 94105, USA
matpol@amazon.com

## Abstract

Many real-world black-box optimization problems have multiple conflicting objectives. Rather than attempting to approximate the entire set of Pareto-optimal solutions, interactive preference learning, i.e., optimization with a decision maker in the loop, allows us to focus the search on the most relevant subset. However, few previous studies have exploited the fact that utility functions are typically monotonic. In this paper, we address the Bayesian Optimization with Preference Exploration (BOPE) problem and propose using a neural network ensemble as a utility surrogate model. This approach naturally integrates monotonicity and allows learning the decision maker's preferences from pairwise comparisons. Our experiments demonstrate that the proposed method outperforms state-of-the-art approaches and is robust to noise in utility evaluations. An ablation study highlights the critical role of monotonicity in enhancing performance.

## 1 Introduction

Global optimization of complex and expensive-to-evaluate functions is a fundamental challenge in various scientific and engineering fields. Bayesian Optimization (BO) has emerged as a powerful framework for addressing such problems in a sample-efficient way. BO has also been extended to multi-objective optimization (MOO), where multiple, often conflicting, objectives need to be optimized simultaneously. It has demonstrated its effectiveness in real-world applications such as protein design, where both protein stability and additional desired properties like solvent-accessible surface area need to be considered, see, e.g., Stanton et al. [2022] or Fromer and Coley [2023].

In MOO, there is usually not a single optimal solution, but a range of so-called Pareto optimal or non-dominated solutions with different trade-offs. Several approaches have been developed for finding good representations of Pareto-optimal solutions, with two methods standing out: ParEGO [Knowles, 2006], which employs random augmented Chebyshev scalarizations for optimization in each iteration, and expected hypervolume maximization [Yang et al., 2019, Daulton et al., 2020], which directly maximizes the hypervolume of the Pareto front. Although both methods can generate a Pareto front (a set of Pareto optimal trade-off solutions), eventually one is usually interested in a single solution, and while it may be convenient for a decision maker (DM) to pick a solution from the Pareto front, it means that a lot of computational effort is spent searching for solutions that are eventually discarded.

---

[*]This research does not relate to Matthias' work at Amazon.

39th Conference on Neural Information Processing Systems (NeurIPS 2025).

To make optimization more efficient, interactive multi-objective optimization algorithms have been proposed that repeatedly elicit preference information from the DM, and then guide the search towards the most preferred solution. Combining expensive MOO and interactive preference learning, Lin et al. [2022] proposed a problem setting called Bayesian Optimization with Preference Exploration (BOPE). In BOPE, the objective values are from an expensive black-box multi-output function $f_{\text{true}}$ called an output function, while the DM preferences are determined by an unknown utility function $g_{\text{true}}$. We observe a solution's objective function values by evaluating the expensive black box function $f_{\text{true}}$. The DM preference is usually elicited through pairwise comparisons between alternatives, as this is known to have low cognitive burden [Guo and Sanner, 2010]. The ultimate goal of BOPE is to efficiently identify the optimal solution $x^*$ that maximizes the DM's utility over the outputs, i.e. $x^* \in \text{argmax}_{x \in \mathcal{X}} g_{\text{true}} \left( f_{\text{true}}(x) \right)$, where $x$ is the decision variable and $\mathcal{X} \subset \mathbb{R}^d$ is the design space. In the original BOPE paper [Lin et al., 2022], a Gaussian process (GP) is utilized to model the utility function given pairwise preference information.

However, a fundamental property of utility functions in conventional economic theory is *monotonicity*, as increasing objective function values should not decrease the utility [Mas-Colell et al., 1995]. Existing approaches usually do not account for monotonicity, potentially hindering the performance of BO. To address this limitation, in this paper, we introduce the *Monotonic Neural Network Ensemble* (MoNNE), a neural network-based approach to modelling utility functions in BOPE. The proposed approach offers several advantages: it naturally incorporates monotonicity constraints through positive weight transformations, efficiently handles pairwise comparison training data using Hinge loss, and quantifies uncertainty through ensemble methods. We also adjust the acquisition function Expected Utility of the Best Option (EUBO) [Lin et al., 2022] that chooses the pairwise comparison for MoNNE models to make it more robust under noise. The experimental results demonstrate that the proposed combination of techniques effectively learns utility functions in BOPE, significantly enhancing performance in multi-objective settings that incorporate DM preferences.

This work makes two key contributions. First, we introduce a model that accounts for monotonicity of utility functions, which achieves better performance over SOTA algorithms. Second, we provide the first comprehensive investigation of neural networks with uncertainty estimation in pairwise comparison settings. Although neural networks have been used in preference learning with pairwise comparison data, uncertainty modelling in this context remains understudied. We compare the commonly used techniques such as Bayesian Neural Networks (BNN) using Hamiltonian Monte Carlo, BNN by Variational Inference, and ensembles. Our experiments reveal that although Bayesian Neural Networks using Hamiltonian Monte Carlo perform well with standard $(X, Y)$ observations, they face significant challenges in pairwise comparisons. As a result, we recommend and validate a neural network ensemble approach as a better alternative to preference learning.

The structure of this paper is as follows. Section 2 surveys the literature on MOO with interactive preference learning. Section 3 introduces the BOPE problem setting. Section 4 presents the neural-network-based modelling approach for the DM's utility as well as the adjusted acquisition function. Section 5 reports on the experimental results, demonstrating the empirical advantage of using MoNNE as the utility surrogate model in BOPE. The paper concludes with a summary and suggests avenues for future work. The code is available at https://github.com/HanyangHenry-Wang/BOPE-MoNNE.git.

## 2 Literature Review

This paper focuses on Bayesian Optimization with Preference Exploration (BOPE), which falls within the broader domain of expensive MOO with interactive preference learning. One common assumption in this domain is that the DM's preference can be modelled by an underlying utility function. Typically, this function is unknown, thus a crucial step is to acquire knowledge of this utility function through pairwise comparisons [Fürnkranz and Hüllermeier, 2010]. Astudillo and Frazier [2020] assume that the utility function family is known and only its parameters are unknown. They utilize Bayesian estimation to learn the parameters and subsequently determine the next solution $x$ to be evaluated, based on an acquisition function called Expected Improvement under Utility Uncertainty (EIUU). Ungredda and Branke [2023] also assume a known utility family but unknown parameters. However, the assumption of knowing the utility function family might be too strong in real life. In the work of Lin et al. [2022], it is assumed that the utility function is entirely unknown, and a GP is employed to model it [Chu and Ghahramani, 2005]. They also extend EIUU to a batched setting, proposing an acquisition function called qNEIUU, and propose an acquisition function called

Expected Utility of the Best Option (EUBO) to select the pair of alternatives to be shown to the DM. Besides GPs, there are other surrogate models. For example, Sinha et al. [2010] use a generalized polynomial value function and point estimation for the model parameters. However, both studies ignore the monotonicity of the utility: increasing values of objective functions should not decrease the DM's utility. To fill this gap, Chin et al. [2018] consider GPs with monotonicity in preference learning. They combined a linear model and a GP as the surrogate model. However, it requires known kernel hyperparameters instead of learning them. There is another possible model called constrained GP (cGP) by López-Lopera et al. [2022]. cGP adapts finite-dimensional GPs (a piece-wise linear approximation of GPs), and then trains the finite-dimensional GPs with constraints on the order of knot values. Barnett et al. [2024] propose a monotonic GP based on cGP. The main difference is that Barnett et al. [2024]'s method does not treat monotonicity as a constraint on the order; instead, they prescribe a monotonic transformation on the knot value. Although these two approaches explicitly model monotonicity, it is not obvious how to use them given pairwise comparison information. In addition, both methods adopt an additive kernel, limiting expressiveness for multiplicative effects in utility functions, such as the Cobb–Douglas utility function. Another approach to incorporate monotonicity is through monotonic parametric utility models such as the Chebyshev scalarization function, as demonstrated by [Ozaki et al., 2024]. However, these methods sacrifice flexibility to achieve monotonicity, and the value of this trade-off is not clear. Concurrent work by Huber et al. [2025] also addresses monotonicity by augmenting the training data with virtual queries in which dominated solutions are labeled as less preferred. However, their approach encourages rather than enforces monotonicity in the model.

A similar problem setting is Preferential Bayesian Optimization (PBO) [González et al., 2017]. PBO aims at optimizing a single black-box function $f(x)$ given pairwise comparison observations. González et al. [2017] use a GP model built on the space of pairs of designs $\mathcal{X} \times \mathcal{X}$ as the surrogate model and propose two acquisition functions, Copeland Expected Improvement and Dueling-Thompson sampling. Subsequent research has built upon and enhanced the PBO framework. Takeno et al. [2023] adapt skewed Gaussian processes [Benavoli et al., 2020] as a surrogate model to improve performance. Furthermore, Siivola et al. [2021] expand PBO to a batched setting, allowing parallel preference queries and potentially accelerating the optimization process. Besides, Astudillo et al. [2023] extend EUBO to the batched PBO setting. There are other variants of PBO, such as amortized PBO [Zhang et al., 2025], projective PBO [Mikkola et al., 2020], and multi-objective PBO [Astudillo et al., 2024]. Recent work by Xu et al. [2024] provides theoretical analysis of PBO, demonstrating that preference-based BO can achieve information-theoretic regret bounds. Furthermore, Arun Kumar et al. [2024] show that PBO can be used to accelerate standard BO by incorporating expert knowledge about abstract, unmeasurable properties. Those later variants no longer build GP models over the space of pairs of designs $\mathcal{X} \times \mathcal{X}$ but directly in the (single) design space $\mathcal{X}$. It should be noted that while PBO and BOPE share conceptual similarities, they differ fundamentally in the type of comparisons being made – a comparison of designs based on decision variables in case of PBO, and based on objective function values in case of BOPE, though PBO can be applied in the BOPE setting if one assumes that the DM can indeed compare sets of decision variables (which may not be true – a customer may be able to sensibly compare two cars based on their observed features such as speed, power, fuel consumption, but not based on their abstract design variables). However, given that both of them use pairwise comparisons, PBO [González et al., 2017, Astudillo et al., 2023] with modification will serve as a benchmark algorithm in our experimental evaluations.

In this paper, we use neural networks as the utility surrogate model due to their flexibility and ability to handle complex, high-dimensional spaces [Snoek et al., 2015, Springenberg et al., 2016, White et al., 2021, Li et al., 2024]. When the number of parameters is large, neural networks can capture intricate patterns in the objective function landscape without making strong assumptions about its underlying structure. In the BO framework, a surrogate model needs to provide both predicted mean and uncertainty estimates. Neural networks offer various approaches to quantifying uncertainty, each with its own strengths and weaknesses. Snoek et al. [2015] introduce the use of Bayesian linear layers on top of deep neural networks, allowing efficient uncertainty estimation while maintaining the expressive power of deep architectures. Springenberg et al. [2016], Kim et al. [2021] and Li et al. [2024] propose a fully Bayesian treatment of neural network parameters, enabling a more comprehensive uncertainty quantification at the cost of increased computational complexity. White et al. [2021] demonstrate the effectiveness of neural network ensembles in capturing predictive uncertainty for Neural Architecture Search, offering a balance between computational efficiency and robust uncertainty estimates. In this paper, we extend the application of neural network ensembles

to BOPE, where the training set consists of pairwise comparisons. Additionally, we enforce model monotonicity to align with the properties of utility functions.

# 3   Problem Setting

Given $k$ different objective functions represented by an expensive black-box multi-output function $f_{\text{true}} : \mathbb{R}^d \to \mathbb{R}^k$. Furthermore, assume that there is a decision maker (DM) whose utility of a solution depends only on these $k$ objective values and follows an unknown utility function $g_{\text{true}} : \mathbb{R}^k \to \mathbb{R}$. The aim is to find

$$x^* \in \text{argmax}_{x \in \mathcal{X}} \{g_{\text{true}} \left(f_{\text{true}}(x)\right)\},$$

where $x$ is the decision variable and $\mathcal{X} \subset \mathbb{R}^d$ is the design space.

The solution process consists of two alternating stages: *Experimentation* and *Preference Exploration*. In the $t$-th iteration:

**Experimentation.** The experimenter selects an $x_t$ for evaluation and observes the values of the objective function $f_{\text{true}}(x_t)$. Given $n$ observations of the objective function values, we denote the set of observations by $\mathcal{D}_n := \{(x_1, y_1), ..., (x_n, y_n)\}$.

**Preference Exploration.** The experimenter asks the DM to express a preference between two outputs, $y_{1,t}$ and $y_{2,t}$, chosen by the experimenter. The DM response is coded as $+1$ if $y_{1,t}$ is preferred and $-1$ if $y_{2,t}$ is preferred. At the $t$-th iteration, when $m$ pairwise comparisons have been observed, the pairwise comparisons are denoted by $\mathcal{P}_t := \{(y_{1,1}, y_{2,1}), ..., (y_{1,m}, y_{2,m})\}$ and $\mathcal{R}_t := \{p_1, p_2, ..., p_m\}$ where $p_i = \pm 1$ is the preference result.

The workflow is illustrated in Figure 1. It supports batch acquisition by selecting multiple solutions $x_t$ and comparison pairs.

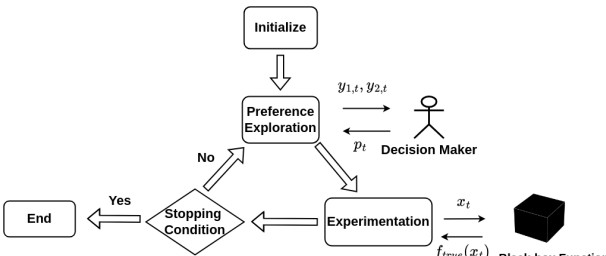

Figure 1: The experiment alternates between the Preference Exploration and the Experimentation stages.

In prior work, Lin et al. [2022] assume that during the Preference Exploration stage, experimenters could present DMs with arbitrary output pairs, including those that were unobserved or even non-existent. However, presenting such unattainable solutions risks causing disappointment when DMs later discover these options are unavailable. Therefore, our work constrains experimenters to only present output pairs that have been previously observed.

In addition, to use a more realistic model of DMs, we allow for noise in the decision process. Lin et al. [2022] discuss two types of noise: (1) a constant probability of the DM making an error, and (2) noise in the utility values. We argue that the first approach oversimplifies real-world decision-making, as errors are often related to the utility gap between options. For example, when the utilities of outputs $y_{1,t}$ and $y_{2,t}$ are very close, the DM is highly likely to make a mistake, but when the utility gap is large, the DM can easily tell which alternative is preferred. Specifically, we assume the decision result is $p_t = 2 \cdot \mathbb{1}_{g_{\text{true}}(y_{1,t}) - g_{\text{true}}(y_{2,t}) + \epsilon > 0} - 1$, where the noise $\epsilon \sim \mathcal{N}(0, \sigma_{\text{noise}}^2)$ for some variance $\sigma_{\text{noise}}^2$. We call noise of this type 'utility noise' in this paper. In addition to the two noise types discussed above, a third type of noise can be introduced by the Bradley-Terry model. Since the noise characteristics of the Bradley-Terry model are similar to that of Gaussian noise, we do not conduct separate experiments with the Bradley-Terry model, but we provide a brief introduction to the Bradley-Terry model and compare it with our Gaussian noise model in Appendix A.5.

# 4 Utility Surrogate Model and Pairwise Comparison Acquisition Function

The problem described above was originally proposed by Lin et al. [2022], and they handle it in a BO framework. This paper follows their framework (albeit with different assumptions about permissible outputs for the Preference Exploration stage) but adjusts the surrogate model and the corresponding acquisition function.

Specifically, their BO framework consists of two surrogate models: a GP model $f$ given $\mathcal{D}_n$ for the objective function, and another GP model $g$ given $\mathcal{P}_m$ and $\mathcal{R}_m$ as the surrogate model for the DM's utility function.

In the Experimentation stage, using these two surrogate models, they select the next evaluation point based on an acquisition function qNEIUU. For a batch of $q$ points $x_{1:q} = (x_1, \ldots, x_q) \in \mathcal{X}^q$, qNEIUU is expressed as: $\text{qNEIUU}(x_{1:q}) = \mathbb{E}_{m,n}\left[\{\max g(f(x_{1:q})) - \max g(f(X_n))\}^+\right]$, where $\mathbb{E}_{m,n}[\cdot] = \mathbb{E}[\cdot \mid \mathcal{P}_m, \mathcal{R}_m, \mathcal{D}_n]$ denotes the conditional expectation given the data from $m$ queries and $n$ experiments, $\{\cdot\}^+$ denotes the positive part function, and following their notation, $\max g(f(x_{1:q})) = \max_{i=1,\ldots,q} g(f(x_i))$ and $\max g(f(X_n)) = \max_{(x,y)\in\mathcal{D}_n} g(f(x))$.

In the Preference Exploration stage, the output pair is chosen according to an acquisition function called the Expected Utility of the Best Option (EUBO) [Lin et al., 2022]. EUBO is defined as $\text{EUBO}(y_{1,t}, y_{2,t}) = \mathbb{E}_m[\max\{g(y_{1,t}), g(y_{2,t})\}]$, where the expectation is over the posterior of the utility surrogate model $g$ at the time the query is chosen. Lin et al. [2022] assume that the experimenter can show the DM any pair for their preference decision even if the solutions do not exist. In this case, it is not straightforward to determine the pair using EUBO because in the acquisition optimization, the search range of the output $y_{1,t}, y_{2,t}$ is unknown. Hence, to find the pair of outputs, EUBO needs to be transformed as $\text{EUBO}(x_{1,t}, x_{2,t}) = \mathbb{E}_m[\max\{g(f(x_{1,t})), g(f(x_{2,t}))\}]$ to find the pair $(x_{1,t}, x_{2,t}) \in \mathcal{X}^2$, and the experimenter can use a posterior sample of $(f(x_{1,t}), f(x_{2,t}))$ as the comparison pair $(y_{1,t}, y_{2,t})$.

In this paper, we follow the steps outlined above. However, we propose using a new surrogate model for the utility function instead of using GPs, as GPs cannot easily incorporate monotonicity constraints. In addition, we adjust the EUBO acquisition function for the new model so that the pairwise comparison can be chosen more efficiently.

## 4.1 Monotonic Neural Network Ensemble as Utility Surrogate Model

An ideal utility surrogate model $g$ for this problem setting should possess the following properties: 1. sufficient flexibility to model realistic utility functions; 2. monotonicity; 3. trained using pairwise comparisons (rankings can also be expressed as a set of pairwise comparisons); 4. capable of handling uncertainty.

We propose using a neural network as the surrogate model. The advantages of neural networks are manifold. First, neural networks with sufficient capacity can approximate complex utility functions without requiring explicit assumptions about their functional form, providing the desired modelling flexibility. Second, by transforming the weights into positive space, we ensure the model is monotonically increasing, as suggested by Sill [1997]. Specifically, we will apply an exponential transformation to the weights, guaranteeing that all weights are positive. The following theorem states that a monotonic neural network is able to model any bounded monotonically increasing function with bounded partial derivatives:

**Theorem 1.** (Sill [1997]) *Let $m(x)$ be any continuous, bounded monotonic function with bounded partial derivatives, mapping $[0,1]^d$ to $\mathbb{R}$. Then there exists a function $m_{net}(x)$ which can be implemented by a monotonic network and is such that, for any $\epsilon$ and any $x \in [0,1]^d$, $|m(x) - m_{net}(x)| < \epsilon$.*

Third, using Hinge loss as the loss function enables us to use pairwise comparisons for training. While Hinge loss is typically used in classification problems, it can also be effectively applied to ordinal regression [Rennie and Srebro, 2005]. Hinge loss is defined as

$$\mathcal{L}(\mathcal{P}_t) = \sum_{(y_{1,i}, y_{2,i})\in\mathcal{P}_t} \max\{0, 1 - \alpha \cdot (g(y_{1,i}) - g(y_{2,i})) \cdot p_i\},$$

where $p_i = \pm 1$ is the corresponding preference of $y_{1,i}$ vs. $y_{2,i}$, and $\alpha$ is a positive parameter that is learned during training. Additionally, while the BOPE problem setting primarily deals with binary

preferences ($\pm 1$), we can extend it to handle tied preferences (denoted by $p_i = 0$). The Hinge loss can be modified accordingly: $\mathcal{L}(\mathcal{P}_t) = \sum_{(y_{1,i}, y_{2,i}) \in \mathcal{P}_t} [\mathbb{1}_{p_i \neq 0} \cdot \max\{0, 1 - \alpha \cdot (g(y_{1,i}) - g(y_{2,i})) \cdot p_i\} + \mathbb{1}_{p_i = 0} \cdot |g(y_{1,i}) - g(y_{2,i})|]$. Lastly, we employ neural network ensembles [Hansen and Salamon, 1990, Lakshminarayanan et al., 2017] to obtain uncertainty estimates that are important for the acquisition criterion.

Combining neural networks, monotonicity constraints, Hinge loss, and ensemble methods as described above, we call the resulting model *Monotonic Neural Network Ensemble* (MoNNE).

To incorporate uncertainty quantification, we explored several established methods, primarily focusing on Bayesian Neural Networks (BNNs) and ensemble techniques. While BNNs with posterior inference performed via Hamiltonian Monte Carlo (HMC) are generally superior for standard supervised learning with direct input-output observations $(X, Y)$ and performs well in 'vanilla' BO (see Figure 5 in Appendix A.2), it encounters significant challenges in preference learning. Briefly, the independent prior distribution of parameters hinders training due to scale invariance in preference learning (because the training set only shows relative comparisons of function values rather than absolute values). We refer to Appendix A.3 for more details. Also, given that Monte Carlo Dropout underperforms compared to ensemble methods [Lakshminarayanan et al., 2017], we focus the investigation on ensemble approaches and BNNs trained with variational inference (VI), particularly recommending ensemble methods for uncertainty quantification in preference learning. This recommendation is supported by several observations: recent studies have shown that ensembles provide more reliable uncertainty estimates than BNNs [Lakshminarayanan et al., 2017, Ovadia et al., 2019]. In the specific context of BO for Neural Architecture Search, White et al. [2021] found ensemble methods to outperform Bayesian estimation approaches, and generally ensembles also show a comparable but marginally lower performance than HMC-trained BNNs in 'vanilla' BO [Li et al., 2024]. Finally, ensembles offer practical advantages through their inherent parallelizability, substantially reducing computational overhead, and as we will demonstrate later, naturally can be combined with normalization to handle scale invariance in preference learning.

In a neural network ensemble, $M_e$ ($M_e > 1$) base neural networks, denoted as $g^j$ ($j = 1, ..., M_e$), are simultaneously trained with the same training set but from different initial parameters. The predicted mean and variance are derived by aggregating outputs from these $M_e$ neural networks. It should be noticed that the output scales of the $M_e$ models might be different due to scale invariance. To handle this issue, we normalize neural networks to ensure uniform scaling. Specifically, the output $g^j(x)$ is scaled as $\frac{g^j(x) - g^j_{\max}}{\sigma^j}$, where $g^j_{\max}$ represents the largest estimated utility value of the compared outputs: $g^j_{\max} = \max_{y \in \{y_i | (y_{1,i}, y_{2,i}) \in \mathcal{P}_m, y_i \in \{y_{1,i}, y_{2,i}\}} \{g^j(y)\}$ and $\sigma^j$ denotes the normalized deviation of compared outputs: $\sigma^j = \sqrt{\frac{\sum_{y \in \{y_i | (y_{1,i}, y_{2,i}) \in \mathcal{P}_m, y_i \in \{y_{1,i}, y_{2,i}\}} (g^j(y) - \mu^j)^2}{|\mathcal{P}_m|}}$, where $\mu^j$ denotes the average predicted utility of model $g^j$ over all outputs in $\mathcal{P}_m$.

Figure 2 shows two surrogate models trained with the same data: a GP and our MoNNE (detailed descriptions of these models are provided in Section 5). The left-hand side of Figure 2 displays two distinct 2D utility functions. The first utility (top) is a linear utility function defined as $g_{\text{true}}(y_1, y_2) = y_1 + 1.5y_2$ and the second (bottom) is more complicated, defined as $g_{\text{true}}(y_1, y_2) = \sqrt{y_1} + 0.9\sin(y_1) + \ln(y_2 + e^{y_1}) + 4.5\sqrt{y_2}$. Both functions increase monotonically in the range $[0, 10]^2$. For clearer visualization, we present slices of these functions. The $x$-axis represents either $y_1$ (value of the first dimension) or $y_2$ (value of the second dimension), as indicated in the legend. For the '$y_1 + y_2 = 10$' slice, the $x$-axis shows $y_1$ values while $y_2 = 10 - y_1$. For '$y_2 = 2.5$', the $x$-axis represents $y_1$ values with $y_2$ fixed at 2.5. Similarly, for '$y_1 = 2.5$', the $x$-axis shows $y_2$ values with $y_1$ fixed at 2.5. Using 15 randomly chosen pairs from $y_1 + y_2 = 10$, we constructed the GP and MoNNE models. The middle plot illustrates the GP surrogate model, while the right plot depicts the MoNNE surrogate model. As evident in Figure 2, the MoNNE model more accurately captures the underlying utility function with only 15 output pairs. Furthermore, by incorporating monotonicity constraints, MoNNE maintains a strictly increasing relationship for legends '$y_1 = 2.5$' and '$y_2 = 2.5$', whereas the GP model exhibits non-monotonic behavior.

Additional visualizations of GPs and MoNNEs with varying numbers of pairwise comparisons are provided in Figure 10 in Appendix A.4.

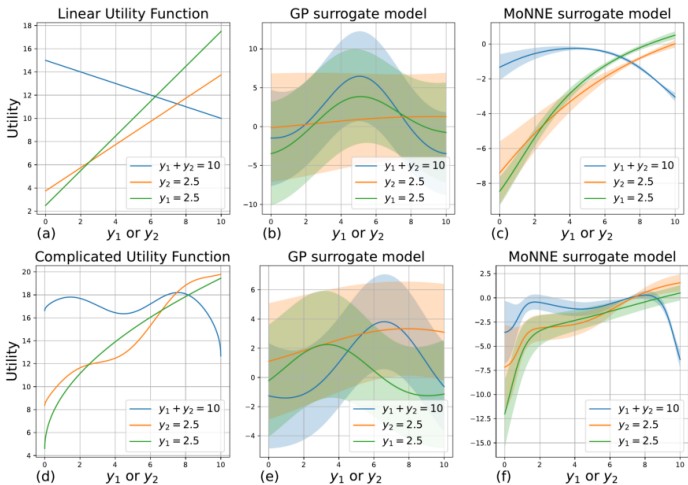

Figure 2: In each row, the left plot depicts the true utility function slices; the middle is the GP surrogate model and the right one is the MoNNE model. While the scale varies significantly across models, this does not affect our analysis, as we care about relative rankings rather than absolute values.

## 4.2 Acquisition Functions

### 4.2.1 Experimentation

After building the two surrogate models $f$ and $g$, we adopt qNEIUU [Astudillo and Frazier, 2020, Lin et al., 2022] to choose the next solution to be evaluated for the Experimentation stage. Following Lin et al. [2022], we calculate the qNEIUU by Monte Carlo simulation:

$$\text{qNEIUU}\,(x_{1:q}) = \frac{1}{M}\frac{1}{N}\sum_{j=1}^{M}\sum_{k=1}^{N}\left\{\max g^j\left(f^k\left(x_{1:q}\right)\right) - \max g^j\left(f^k\left(X_n\right)\right)\right\}^{+},$$

where $x_{1:q} \in \mathcal{X}^q$ (in this paper, $q$ is set to be 1), $X_n$ is the tensor of evaluated solutions, $\{\cdot\}^+$ denotes the positive part function, $M$ is the number of samples from the model $g$, $N$ is the posterior sample number of $f$, $g^j(\cdot)$ is the $j$-th sample from $g$, and $f^k(\cdot)$ is the $k$-th posterior sample from $f$. When the utility surrogate model $g$ is a MoNNE, we set $M = M_e$.

### 4.2.2 Preference Exploration

For the Preference Exploration stage, EUBO can be used to pick output pairs. As mentioned, in this paper, the experimenter is only allowed to show observed outputs. In this case, EUBO can be adjusted as $\text{EUBO}\,(y_{1,i}, y_{2,i}) = \frac{1}{M}\sum_{j=1}^{M}\max\left\{g^j\left(y_{1,i}\right), g^j\left(y_{2,i}\right)\right\}$, where $y_{1,i}, y_{2,i} \in \{y_i | (x_i, y_i) \in \mathcal{D}_n\}$ are two observed outputs.

In this paper, we propose one modification to the EUBO for MoNNEs. Specifically, when calculating $\max(g(y_{1,t}), g(y_{2,t}))$, we assume independence between $g(y_{1,t})$ and $g(y_{2,t})$, instead of treating them as correlated. We term the modified acquisition function Independent EUBO (IEUBO). Specifically, IEUBO is calculated as:

$$\text{IEUBO}\,(y_{1,t}, y_{2,t}) = \mathbb{E}[\max\left\{g\left(y_{1,t}\right), g\left(y_{2,t}\right)\right\}],$$

where $y_{1,t}, y_{2,t} \in \{y_i | (x_i, y_i) \in \mathcal{D}_n\}$ are two observed outputs, and $g(y_{1,t})$ and $g(y_{2,t})$ are two independent Gaussian variables predicted by the surrogate model $g$. The expectation can be computed analytically: for two independent normal random variables $X \sim \mathcal{N}(\mu_1, \sigma_1^2)$ and $Y \sim \mathcal{N}(\mu_2, \sigma_2^2)$, we have $\mathbb{E}[\max(X, Y)] = \mu_1 \Phi(\alpha) + \mu_2 \Phi(-\alpha) + \sigma_3 \phi(\alpha)$, where $\Phi$ is the cumulative distribution function of a standard normal distribution, $\sigma_3 = \sqrt{\sigma_1^2 + \sigma_2^2}$ and $\alpha = \frac{\mu_1 - \mu_2}{\sigma_3}$ [Nadarajah and Kotz, 2008]. The derivation details can be found in Appendix A.10.

Our experiments demonstrate that IEUBO generally outperforms EUBO in the presence of utility noise. Detailed experimental results are provided in Appendix A.8. This performance difference may be attributed to how noise affects DM preferences: under noisy conditions, the base model of MoNNE might learn incorrect correlations between outputs, potentially making the independence assumption of IEUBO more robust than EUBO's consideration of correlations. Hence, we use IEUBO as the acquisition function.

The use of independence has precedent: White et al. [2021] employed independent Thompson Sampling for neural network ensemble surrogate model in the Neural Architecture Search (NAS) problem, where they similarly assumed that posterior distributions are independent.

Note that GP models do not suffer from this unreliable correlation issue. Therefore, when using GP utility surrogate models, we still employ EUBO for selecting output pairs to present to the DM, albeit constrained to the observed outputs.

Algorithm 1 in Appendix A.1 summarizes the BOPE algorithm using MoNNE as the utility surrogate model and IEUBO to decide the output pairs for comparison.

## 5 Experimental Results

In this section, we compare the proposed method with benchmark algorithms on six test problems. The performance of different algorithms is measured by simple regret, defined as the gap between the true optimal utility and the best utility over the observations: $\max_{x \in \mathcal{X}} \{g_{\text{true}}(f_{\text{true}}(x))\} - \max_{x \in \{x_i | (x_i, y_i) \in \mathcal{D}_n\}} \{g_{\text{true}}(f_{\text{true}}(x))\}$.

We use all four multi-objective problems that were used as benchmarks in [Lin et al., 2022], plus two additional ones. We test a wide range of utility functions from Lin et al. [2022], Astudillo and Frazier [2020], plus the Cobb-Douglas function, which is a standard function in economics, and LinearExponential, a sum of a linear component and an exponential product. Details can be found in Appendix A.5. Following the experiment setting of Lin et al. [2022], the initial number of output observations, that is, the initial dataset, is 16 (for $d < 5$) or 32 (for $d \geq 5$), and the number of initial pairwise comparisons is 10 (for $d < 5$) or 20 (for $d \geq 5$). In each iteration, we evaluate one solution, observing $f_{\text{true}}(x)$ and ask the DM for his preference over one pair of outputs. We compare the proposed method against the following benchmark algorithms: Random, qNParEGO [Daulton et al., 2020], qNEHVI [Daulton et al., 2020], PBO [Astudillo et al., 2023], BOPE-Linear (using a linear model as the utility surrogate model), BOPE-GP [Lin et al., 2022], BOPE-MoNNE (Ours), BOPE-BMNN (Ours, using Variational Inference-trained Bayesian Monotonic Neural Network), Known Utility (the utility function is known, so the problem becomes a Composite BO [Astudillo and Frazier, 2019]). For more details about benchmark algorithms, see Appendix A.5.

We allow for noise in DM's decision, as discussed in Section 3. By default, $\sigma_{\text{noise}}$ is set to $0.1$ (noise is a setting of the test problem, not the algorithm). Although $0.1$ may seem small, considering that BOPE-MoNNE can achieve regret less than $0.001$ as shown in Figure 3, $0.1$ utility noise can cause enough trouble (Figure 21 in Appendix A.9 shows how many preference errors $0.1$ utility noise can make). We also conduct tests with $\sigma_{\text{noise}} = 0$ and $\sigma_{\text{noise}} = 0.5$ to evaluate how this noise affects the performance of MoNNEs and GP surrogate models, and the result is shown in Figure 22 in Appendix A.9.

The MoNNE implementation employs an ensemble of 8 three-layer neural networks. The three layers have dimensions $d \times 100$, $100 \times 10$, $10 \times 1$, where the weights are transformed using an exponential function. This relatively shallow architecture was chosen due to the limited training set size in the BOPE problem setting. For initialization, the network parameters are drawn uniformly random from a range that depends on the number $s$ of inputs to a node. Specifically, weights are sampled from `[-(1/s)-6,1/s]` and biases from `[-1/s,1/s]`. The weight initialization uses a shifted range with a reduced lower bound to account for the subsequent exponential transformation that ensures positive weights and thus monotonicity. In terms of activation function, we utilize Swish [Ramachandran et al., 2017]. A study of the influence of activation functions and layer numbers can be found in Appendix A.6 and A.7. We use the Adam optimizer (lr = 0.01) combined with CosineAnnealingLR (`T_max` = 1600, `eta_min` = 0.0001) for MoNNE training, and the training epoch number is 1600. Also, to save computation time, we finish the training when the Hinge loss value reduces to zero. The acquisition function optimization uses the multi-start L-BFGS method [Balandat et al., 2020], and all

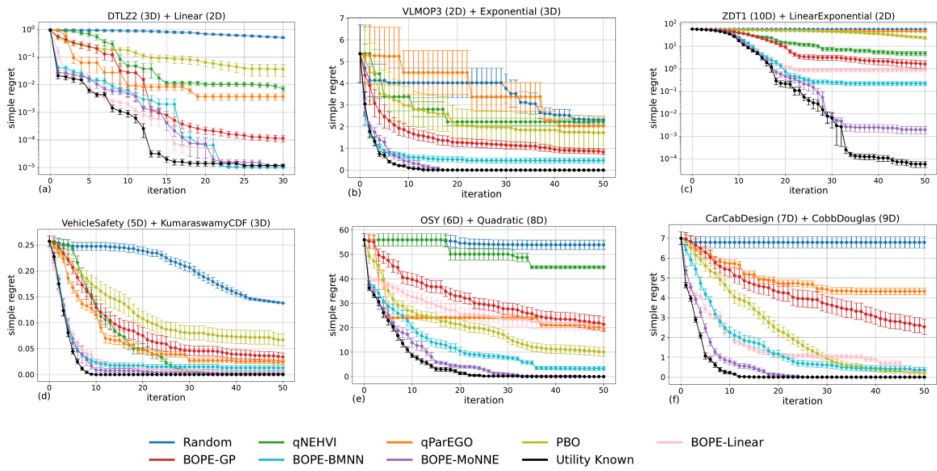

Figure 3: Experimental results show that among all BOPE algorithms, BOPE-MoNNE demonstrates superior performance, followed by BOPE-BMNN and BOPE-GP.

hyperparameters are identical (initial sample number 256, restart number 12) for all algorithms. The output GP surrogate model uses a Matérn kernel with Automatic Relevance Determination (ARD). All experiments are replicated 20 times and the plots show the mean $\pm 1$ standard error. To save computational cost, we stop a run early if the simple regret drops below $10^{-5}$. If this happens, the value for this replication is set to $10^{-5}$ for this and all subsequent iterations.

Figure 3 compares the performance of different methods, with subplots titled 'A+B' (A: output function, B: utility function). Excluding Known Utility algorithm, among all the BOPE algorithms, BOPE-MoNNE consistently achieves the best performance, followed by BOPE-BMNN and BOPE-GP, aligning with the uncertainty quality of these models reported in Table 1 (Appendix A.5). We evaluated uncertainty quality using a modified cross-entropy loss function: $\mathbb{1}_{p=1} \cdot \mathbb{P}(p=1) + \mathbb{1}_{p=-1} \cdot \mathbb{P}(p=-1)$, where $\mathbb{P}(p=1)$ and $\mathbb{P}(p=-1)$ represent the model's predicted probabilities for the DM's preferences. This metric takes into account both the accuracy of the model and its uncertainty, and the larger the value, the better. From Table 1, MoNNE exhibits the highest uncertainty quality, with BNN following and GP ranking last. It should be noticed that the poor uncertainty quality of GPs is mainly due to scale invariance, while the GP prediction accuracy is still good, although lower than MoNNE and BNN (see Table 2 in Appendix A.5). The experiment details about these tables are reported in Appendix A.5. In addition, BOPE-Linear works slightly better than BOPE-MoNNE when the utility function is linear, which is not surprising. However, in the other problems, the MoNNE model consistently outperforms the linear model.

Furthermore, we find that constraining output comparisons to observed values diminishes BOPE-GP's performance (see Figure 19 in Appendix A.8). While removing this constraint improves BOPE-GP's effectiveness, BOPE-GP still underperforms compared to BOPE-MoNNE.

To gain more insight into the effectiveness of the proposed method, we performed an ablation study to isolate the impact of two key components of MoNNE: the monotonicity constraint and the ensemble technique.

Figure 4 illustrates the results (20 repetitions) of the ablation experiments across various test problems, consistently demonstrating that the full BOPE-MoNNE model, incorporating both monotonicity constraints and ensemble learning, outperforms its ablated variants and competing methods (without ensemble can be considered as a case where we use Monte Carlo simulation with only one sample for acquisition function calculation). The inclusion of monotonicity information significantly enhances the model's performance by leveraging domain knowledge about the underlying preference structure. Similarly, the use of ensemble techniques proves beneficial, likely due to its ability to capture a broader range of potential preference patterns and reduced over-fitting. Note that even when the monotonicity constraint is removed, BOPE-MoNNE still outperforms BOPE-GP, indicating that the neural network ensemble provides advantages beyond just its ability to incorporate monotonicity information.

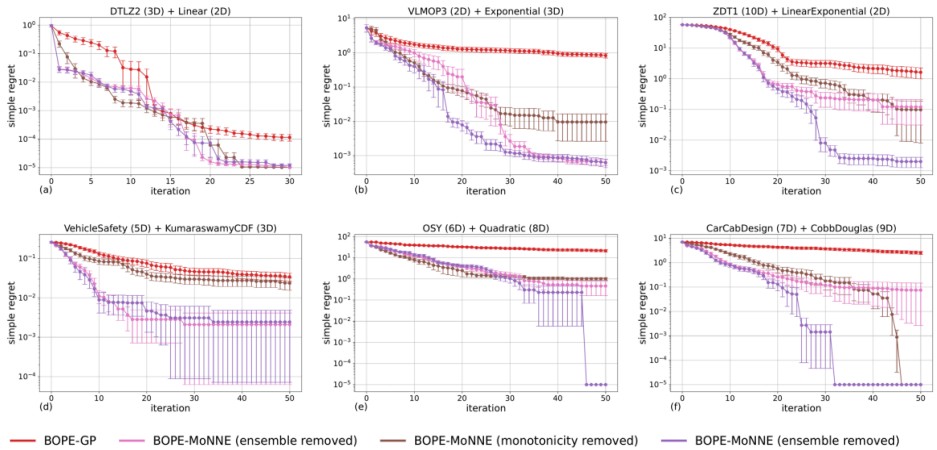

Figure 4: The ablation study shows that MoNNE performs best, highlighting the importance of all components. Regret values capped at $10^{-5}$.

# 6 Conclusion

This paper introduces BOPE-MoNNE (Bayesian Optimization with Preference Exploration using a Monotonic Neural Network Ensemble), a novel approach to multi-objective optimization that learns the decision maker's preferences, leveraging monotonicity of the underlying utilities. The experimental results demonstrate the effectiveness of this method across a range of test problems. BOPE-MoNNE consistently outperforms other methods, including BOPE-GP [Lin et al., 2022] and PBO [Astudillo et al., 2023], across various test problems. A key strength of the proposed approach lies in its ability to incorporate monotonicity of utilities through our neural network design, addressing a crucial property of utility functions that is often overlooked in existing approaches.

Looking ahead, there are several promising directions for future work. This paper identified challenges faced by HMC-trained BNNs in preference learning, particularly due to the scale invariance of preference learning and the independent prior distributions. An important area of exploration is how to effectively utilize HMC to train BNNs given pairwise comparisons, as an alternative to ensemble-based approaches. Additionally, we are interested in extending the work to group decision-making scenarios, where multiple decision makers are involved. Preference-based multi-objective optimization has a wide range of applications, from science over engineering to business problems that we expect to benefit from the advancements presented in this work.

**Acknowledgments**

We sincerely thank the anonymous reviewers for their valuable comments and constructive suggestions, which have greatly helped improve the quality of this paper. The first author gratefully acknowledges support by the Engineering and Physical Sciences Research Council through the Mathematics of Systems II Centre for Doctoral Training at the University of Warwick (reference EP/S022244/1).

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

# A  Appendix

## A.1  Algorithm

---
**Algorithm 1** BOPE-MoNNE

---
1:  Initial observation $\mathcal{D}_0$, initial pairwise comparisons $\mathcal{P}_0$, and $\mathcal{R}_0$
2:  **for** $t = 0, 1, \ldots, T$ **do**
3:      Train a GP output model $f$ given $\mathcal{D}_t$
4:      Train a MoNNE utility model $g$ given $\mathcal{P}_t$ and $\mathcal{R}_t$
5:      **Experimentation Stage:**
6:      Find $x_{t+1} \in \arg\max_{x \in \mathcal{X}} \{\text{qNEIUU}(x|f, g)\}$
7:      Query objective function: $y_{t+1} = f_{\text{true}}(x_{t+1})$
8:      Update $\mathcal{D}_{t+1} = \mathcal{D}_t \cup \{(x_{t+1}, y_{t+1})\}$
9:      **Preference Exploration Stage:**
10:     Find $(y_{1,t+1}, y_{2,t+1}) \in \arg\max_{\{y_{1,i}, y_{2,i}\}} \{\text{IEUBO}(y_{1,i}, y_{2,i}|g)\}$,
11:      where $y_{1,i}, y_{2,i} \in \{y_i | (x_i, y_i) \in \mathcal{D}_n\}$
12:     Query preference: $p_{t+1} = 2 \cdot \mathbb{1}_{g_{\text{true}}(y_{1,t+1}) - g_{\text{true}}(y_{2,t+1}) + \epsilon > 0} - 1,\ \epsilon \sim \mathcal{N}(0, \sigma_{\text{noise}}^2)$
13:     Update $\mathcal{P}_{t+1} = \mathcal{P}_t \cup \{(y_{1,t+1}, y_{2,t+1})\}$ and $\mathcal{R}_{t+1} = \mathcal{R}_t \cup \{p_{t+1}\}$
14: **end for**

---

## A.2  'Vanilla' BO by Neural Network Ensemble and HMC-trained BNN

In this section, we evaluate two surrogate models in BO: neural network ensemble and Bayesian Neural Networks (BNNs) with posterior inference performed via Hamiltonian Monte Carlo (HMC). All neural networks are of 4 layers with size 128 (as suggested by Li et al. [2024]) and Swish as the activation function. For BNN training, we utilize the *pyro* package [Bingham et al., 2018, Phan et al., 2019] with the No-U-Turn Sampler (NUTS) algorithm [Hoffman et al., 2014], a variant of HMC. The NUTS hyperparameters include 50 warmup iterations and 50 posterior samples.

The experiment involves four common synthetic objective functions, with an initial sample size of $3d$ ($d$ being the input dimension) and 10 repetitions. We employ the qEI acquisition function from BoTorch and $q$ is set to be 5. Our results demonstrate that HMC-trained BNNs exhibit performance comparable to ensemble methods, aligning with the findings of Li et al. [2024].

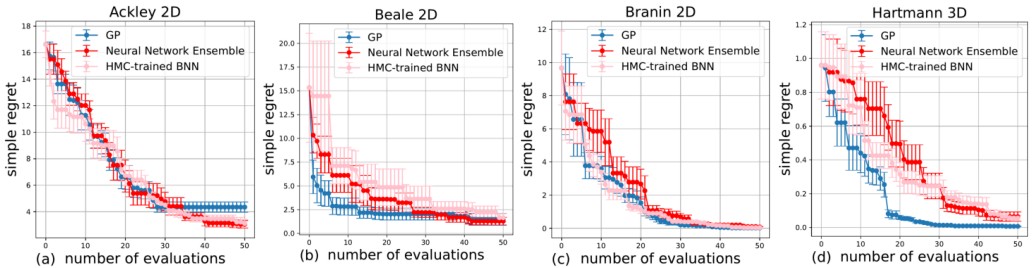

Figure 5: HMC-trained BNNs share similar performance as neural network ensemble in 'vanilla BO' setting.

## A.3  HMC-trained BNN Given Pairwise Comparisons

In this section, we show empirical results demonstrating that HMC does not work well in training BNNs given pairwise comparisons. We attribute this to the scale invariance of preference learning and the independent prior distribution of neural network parameters. Specifically, in preference learning, due to scale invariance, the most important thing to learn is the relationship between parameters rather than their absolute value. However, in Bayesian estimation, the prior distributions of different parameters are independent. Hence, there is a big prior conflict which hinders the training.

To clarify this issue, we first consider the simplest case: a linear utility function $f_{\text{true}}(x_1, x_2) = x_1 + 2x_2$ and a linear model $f(x_1, x_2) = w_1 x_1 + w_2 x_2$. To be simple, we assume no utility noise, and we are given pairwise comparisons. In this case, we do not care about the absolute values of $w_1$ and $w_2$, but rather their relationship, i.e., $\frac{w_1}{w_2} = \frac{1}{2}$.

Using Maximum Likelihood Estimation (MLE), given 20 pairs, we can learn $\frac{w_1}{w_2} \approx \frac{1}{2}$. However, with HMC using standard normal priors for each parameter, we can quickly learn that $\frac{\mu(w_1)}{\mu(w_2)} \approx \frac{1}{2}$ (where $\mu(\cdot)$ denotes the posterior mean), but their estimated correlation $\rho$ is only 0.75, far from the ideal case of $\rho = 1$. This relatively low correlation cannot guarantee $\frac{w_1}{w_2} \approx \frac{1}{2}$ when sampling $w_1$ and $w_2$ from their posterior distribution. Increasing the training size to 40 improves the correlation to 0.85, which is higher but still far from $\rho = 1$. The relationship between correlation and sample size is illustrated in Figure 6. As shown, the correlation approaches 1 as the training set grows, requiring 100 data points to achieve $\rho \approx 0.95$, which is a large number of samples.

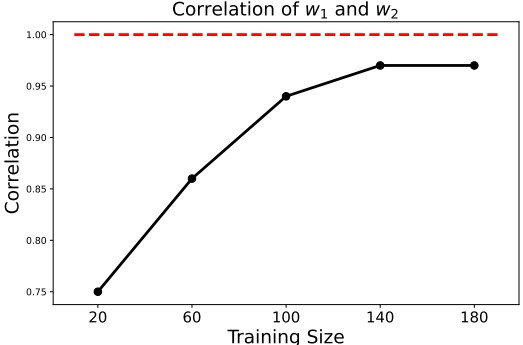

Figure 6: A large number of samples are needed to learn the relationship between $w_1$ and $w_2$.

The key difference between MLE and Bayesian estimation by HMC lies in the independence of the prior: in preference learning, parameters should be highly correlated due to scale invariance, but the prior assumes independence. Consequently, the prior hinders the training, necessitating a much larger training dataset to learn the underlying relationship.

Turning to BNN models, we maintain the setup from Section 5 (but remove monotonicity) with independent standard normal prior distributions for weights and biases. Our goal is to model a one-dimensional function $f_{\text{true}}(x) = \sqrt{x} + 0.42\sin(2x)$ using pairwise comparisons. With 100 randomly sampled comparisons, the model's fitness remains poor (as shown in the Figure 7(a)). Examining the posterior distribution of weights in the last layer reveals a distribution nearly identical to the prior (mean 0, variance 1), indicating that the prior distribution may be too restrictive. By increasing the training set to 2000 samples, the model fits the underlying function well (see Figure 7(b)). At this point, the posterior distributions diverge significantly from the prior. This more complex example reinforces our earlier explanation: scale invariance and the independence prior impede effective training with HMC.

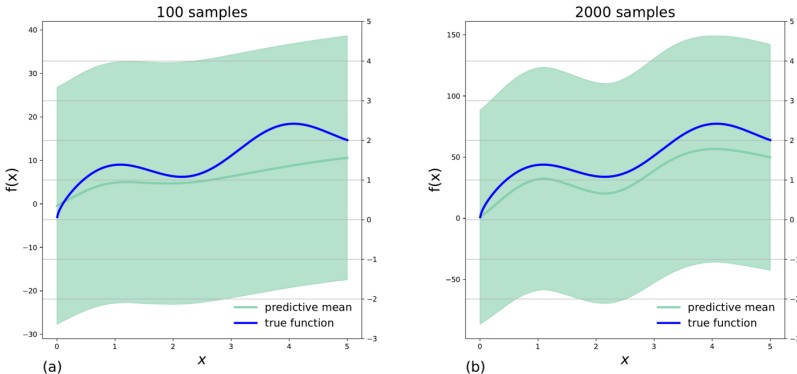

Figure 7: (a) With 100 pairwise comparisons, the BNN model exhibits poor fit; (b) With 2000 data points, the fit improves significantly. Note: The large uncertainty arises from scale invariance.

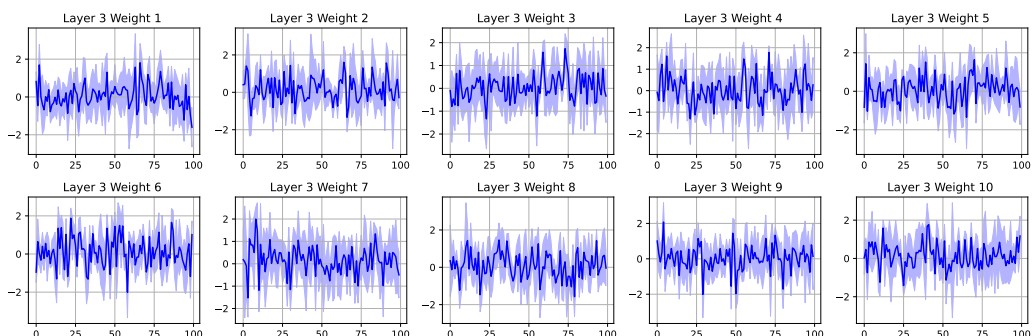

Figure 8: After 100 pairwise comparisons, the posterior distribution of the weights remains nearly indistinguishable from the prior distribution $\mathcal{N}(0, 1)$.

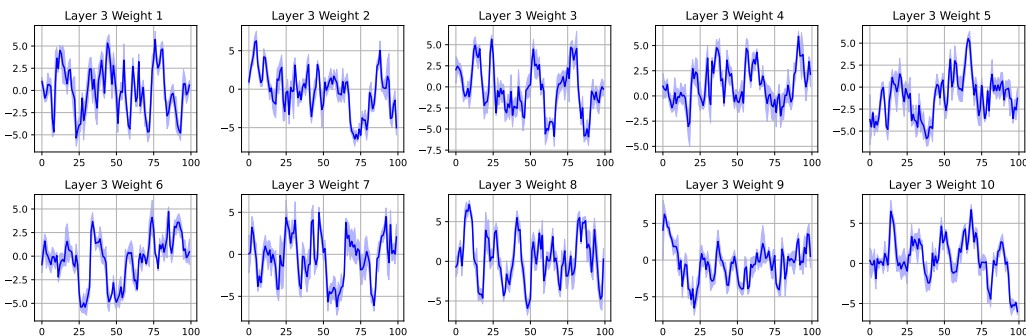

Figure 9: With 2000 pairwise comparisons, the posterior distribution deviates significantly from the prior distribution.

Consequently, HMC-trained BNNs are unsuitable for our problem setting, as they require a large number of training data points. This is particularly challenging in BO problems, where data is limited.

All the experiments above have been executed using the *pyro* package [Bingham et al., 2018, Phan et al., 2019], and we use NUTS [Hoffman et al., 2014] as the HMC algorithm.

### A.4 Model Visualization Given Different Numbers of Pairwise Comparisons

This section presents GP and MoNNE models applied to the complex function described in Section 3, with varying numbers of pairwise comparisons, as illustrated in Figure 10. The top row displays the GP model, while the bottom row showcases the MoNNE model. For GP models, the 'two-peak' pattern for $y_1 + y_2 = 10$ becomes apparent only after 30 samples, whereas for MoNNE, this pattern is evident with just 15 samples.

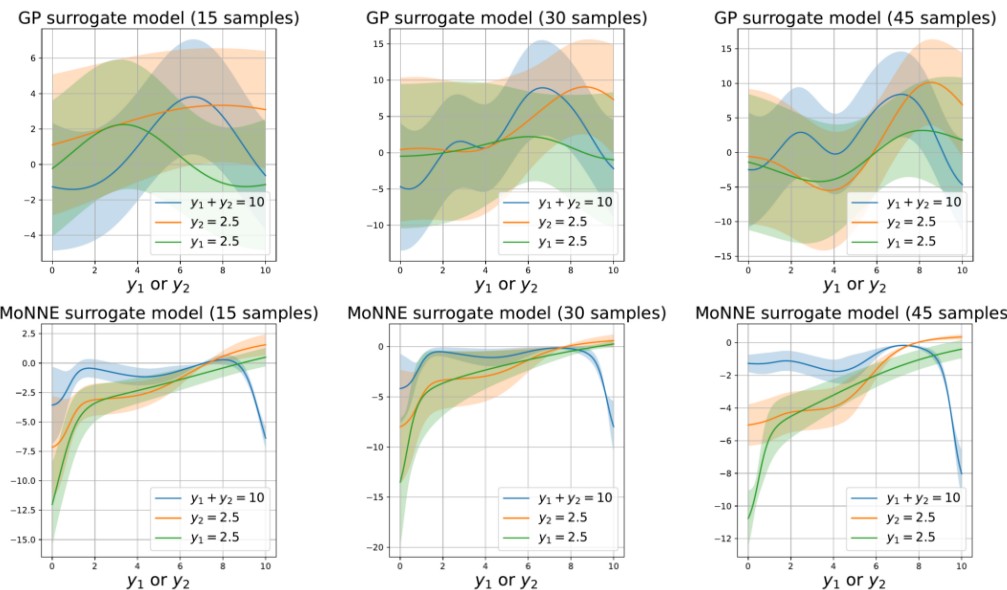

Figure 10: GPs vs MoNNEs given different number of pairwise comparisons

### A.5 Experiment Setting

#### A.5.1 Noise Model

In this paper, we consider noise in utility values, which can lead to errors in the DM's choices. Another type of noise can be introduced through the Bradley-Terry model.

The Bradley-Terry preference model defines the probability of preferring item $i$ over item $j$, given their utility values $U_i$ and $U_j$, as $P(i \succ j) = \frac{e^{U_i}}{e^{U_i} + e^{U_j}} = \frac{1}{1 + e^{-\delta}}$, where $\delta = U_i - U_j$ denotes the utility gap. A fundamental limitation of the standard Bradley-Terry model is its sensitivity to utility scale. For example, if a utility function ranges over $[0, 1]$ and we scale all utility values by 10 to obtain a new function ranging over $[0, 10]$, the predicted probabilities change significantly, even though the preference ordering remains the same. To address this issue, a more flexible formulation introduces a scale parameter $\beta$, yielding $P(i \succ j) = \frac{1}{1 + e^{-\beta \delta}}$. With the additive Gaussian noise used in our experiments, i.e., $\epsilon \sim \mathcal{N}(0, \sigma_{\text{noise}}^2)$, the probability of selecting item $i$ becomes $P(U_i - U_j + \epsilon > 0) = \Phi\left(\frac{\delta}{\sigma_{\text{noise}}}\right)$, where $\Phi(\cdot)$ denotes the standard normal cumulative distribution function.

Both the Bradley-Terry and Gaussian noise models exhibit similar behavior: as $\delta \to 0$, the selection probability approaches 0.5, and as $\delta \to \infty$, it approaches 1. Moreover, for a given noise level $\sigma_{\text{noise}}$, one can choose a corresponding $\beta$ such that the two models closely align. For instance, when $\sigma_{\text{noise}} = 1$, setting $\beta = 1.8$ results in almost identical probability curves (see Figure 11). Therefore, we expect that our results will not change if our Gaussian-noise based formulation would be replaced by the very similar Bradley-Terry choice model.

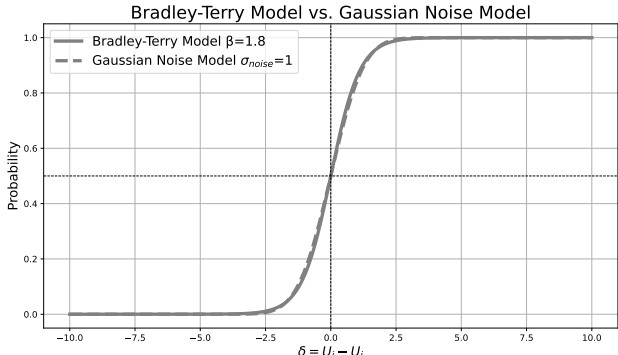

Figure 11: Bradley-Terry and Gaussian noise models exhibit similar behaviour.

### A.5.2 Test Problems

The test output problems are taken from [Lin et al., 2022, Astudillo and Frazier, 2020], combined with different utility functions. An interesting fact is that, in terms of VehicleSafety, OSY, and CarCabDesign, a global optimum $x^*$ is located at the corner of the search range. Hence, although these three problems are of higher dimension, it is relatively easier to find the optimal solution as BO tends to explore more on the boundary [Oh et al., 2018].

- DTLZ2 (3D) + Linear (2D): The output function DTLZ2, introduced by Deb et al. [2005], is a two-objective problem. Its objective functions are defined as $f_{\text{true}}^0(\mathbf{x}) = (1 + h(\mathbf{x})) \cdot \cos(x_0 \cdot \pi/2)$ and $f_{\text{true}}^1(\mathbf{x}) = (1 + h(\mathbf{x})) \cdot \sin(x_0 \cdot \pi/2)$, where $h(\mathbf{x}) = \sum_{i=m}^{d-1}(x_i - 0.5)^2$.

  The utility function is $g_{\text{true}}(\mathbf{y}) = \sum_{i=1}^{k} \theta_i \cdot y_i$, where $\theta = [3.5, 6.5]$.

- VLMOP3 (2D) + Exponential (3D): The output function VLMOP3 [Van Veldhuizen and Lamont, 1999] has three objectives: $f_{\text{true}}^0(\mathbf{x}) = 0.5\left(x_1^2 + x_2^2\right) + \sin\left(x_1^2 + x_2^2\right)$, $f_{\text{true}}^1(\mathbf{x}) = \frac{(3x_1 - 2x_2 + 4)^2}{8} + \frac{(x_1 - x_2 + 1)^2}{27} + 15$ and $f_{\text{true}}^2(\mathbf{x}) = \frac{1}{x_1^2 + x_2^2 + 1} - 1.1 \cdot \exp\left(-\left(x_1^2 + x_2^2\right)\right)$.

  The utility function is $g_{\text{true}}(\mathbf{y}) = \sum_{i=1}^{k} \frac{1 - \exp(-\theta y_i)}{\theta}$, where $\theta = 0.35$.

- ZDT1 (10D) + LinearExponential (2D): The output function ZDT1, introduced by Zitzler et al. [2000], is a two-objective benchmark problem. Its objective functions are defined as $f_{\text{true}}^0(\mathbf{x}) = x_0$ and $f_{\text{true}}^1(\mathbf{x}) = g(\mathbf{x}) \cdot \left(1 - \sqrt{x_0/g(\mathbf{x})}\right)$, where $g(\mathbf{x}) = 1 + \frac{9}{d-1}\sum_{i=1}^{d-1} x_i$.

  The utility function is $g_{\text{true}}(\mathbf{y}) = 5(y_0 + 2) + 5(y_1 + 2) + 2\exp(0.75(y_0 + 2))\exp(1.25(y_1 + 2))$.

- OSY (6D) + Quadratic (8D): The output function, the OSY problem, originally introduced as a constrained optimization task by Osyczka and Kundu [1995], was reformulated as a multi-objective problem by redefining its constraints as objectives.

  The utility function is $g_{\text{true}}(\mathbf{y}) = \sum_{i=1}^{k} y_i^2$.

- VehicleSafety (5D) + KumaraswamyCDF (3D): For details of the output function $f_{\text{true}}$ definition, we refer readers to Liao et al. [2008], Tanabe and Ishibuchi [2020]. In addition, each dimension of $f_{\text{true}}$ is normalized to be within the range $[0, 1]$.

  The utility function is $g_{\text{true}}(\mathbf{y}) = \prod_{i=1}^{k} F_i(y_i)$. where $F_i(\cdot)$ is the CDF of a Kumaraswamy distribution with shape parameters $\mathbf{a} = [0.5, 1, 1.5]$ and $\mathbf{b} = [1.0, 2.0, 3.0]$.

- CarCabDesign (7D) + CobbDouglas (9D): Details can be found in Deb and Jain [2013], Tanabe and Ishibuchi [2020]. To ensure a deterministic ground-truth outcome function, the stochastic components of the original problem were excluded. Each dimension of $f_{\text{true}}$ was also normalized to the range $[0, 1]$ during the simulation.

  The utility function is $g_{\text{true}}(\mathbf{y}) = C \cdot \prod_{i=1}^{k}(y_i + \theta_i)$, where $C = 50$ and $\theta = [0.75, 0.5, 0.5, 0.3, 0.7, 0.5, 0.65, 0.8, 0.55]$.

### A.5.3 Benchmark Algorithms

We compare the proposed method against the following benchmark algorithms:

- Random: A baseline approach that randomly selects the next solution with SobolEngine.

- qNParEGO [Daulton et al., 2020]: A multi-objective optimization algorithm that operates without learning DM preferences. Multi-objective optimization algorithms can be considered as strategies where the experimenter tries to find promising solutions on the Pareto front and asks the DM for their preference at the end.

- qNEHVI [Daulton et al., 2020]: Another multi-objective optimization algorithm that, like qNParEGO, does not incorporate DM preference learning.

- PBO [Lin et al., 2022, Astudillo et al., 2023]: An approach that learns DM preferences directly in the solution space $\mathcal{X}$. It constructs a model $g(x)$ to predict the DM preference for a solution $x$. The acquisition functions are defined as qNEIUU $(x_{1:q}) = \frac{1}{M} \sum_{j=1}^{M} \left\{ \max g^j(x_{1:q}) - \max g^j(X_n)) \right\}^+$ and EUBO $(x_{1,t}, x_{2,t}) = \frac{1}{M} \sum_{j=1}^{M} \max \left\{ g^j(x_{1,i}), g^j(x_{2,i}) \right\}$. It is important to note that we still require an output surrogate model $f$ as DM can only compare the output pair rather than the input pairs. As the true output function is unknown, we use the posterior samples of $(f(x_{1,i}), f(x_{2,i}))$ as output pairs. As mentioned in the literature review, PBO is different from BOPE as their information sources are different. In the BOPE problem setting, DMs compare solutions based on their objective function values, while in PBO, comparisons are based on their design variables, making PBO not directly applicable. Therefore, in our benchmark PBO algorithm implementation, we construct a surrogate model of the output function and use output samples from pairs selected by PBO to elicit DM preferences in the Preference Exploration stage. In the Experimentation stage, we use the Expected Improvement acquisition function to pick a solution to be evaluated. Its result is shown in Figure 3. However, this setting may show the DM any arbitrary output pairs instead of only observed ones. Another way to make PBO applicable is to do two experimentation evaluations combined and then one preference exploration. This approach is necessary for PBO because a DM cannot directly compare two different inputs; instead, we must first obtain the outputs from both methods and subsequently elicit the DM's preferences between these outputs. We conducted experiments using this more equitable framework (one preference exploration and two experimentation evaluations in every iteration). Results presented in Figure 12 show that the order of performance of PBO, BOPE-GP and BOPE-MONNE remains the same.

- BOPE-Linear: Using a linear model of the utility function. Similar to MoNNE, we constrain the parameters of the linear model to be positive to guarantee utility to be monotonically increasing. We use Laplace approximation as an approximation of the Bayesian estimation. This model is monotonically increasing but lacks the flexibility to represent more complex utility functions.

- BOPE-GP [Lin et al., 2022]: A method specifically designed for BOPE problems that models DM preferences using GPs and employs EUBO to select pairwise comparisons from the observed outputs.

- BOPE-MoNNE (Ours): Building upon BOPE-GP's framework, our method replaces the GP utility surrogate model with MoNNE and uses IEUBO for pairwise comparison selection, as detailed in Section 4.

- BOPE-BMNN (Ours): We implemented a Bayesian Monotonic Neural Network (BMNN) as a benchmark surrogate model to evaluate the effectiveness of different uncertainty sources. As discussed, HMC faces some challenges given the pairwise comparison training set, so VI is utilized to train the model. The implementation is based on the *torchbnn* package [Lee et al., 2022]. We define the loss function as a weighted sum of the expected Hinge loss and the KL-divergence, $\mathbb{E}_{\theta \sim q}[\mathcal{L}(\mathcal{P}_t)] + \lambda \cdot D_{KL}(q(\theta)||p(\theta))$, where $\theta$ represents the neural network parameters, i.e. weights and biases, $q(\theta)$ is the posterior distribution of $\theta$, and $p(\theta)$ is its prior. The weight $\lambda$ is set to $10^{-5}$: we use such a small positive value to balance the scales of the KL-divergence and Hinge loss (see visualization of BMNNs with various $\lambda$ values in Figure 13). To ensure comparability with BOPE-MoNNE, we use IEUBO to select output pairs.

- Utility Known: The utility function is known, so the problem becomes a Composite BO [Astudillo and Frazier, 2019]. This algorithm is considered as the potential best benchmark that one can achieve.

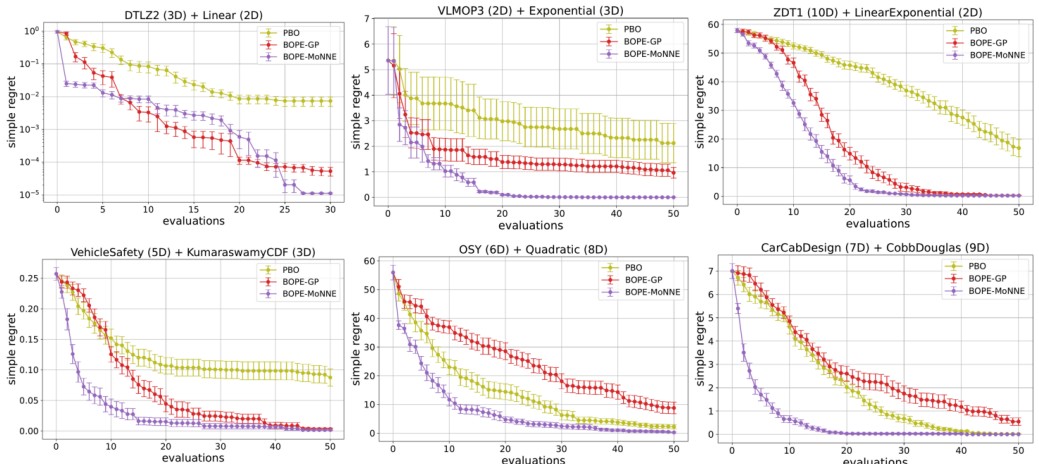

Figure 12: The order of performance of PBO, BOPE-GP and BOPE-MONNE does not change even if we allow two experimental evaluations in every iteration (which is more natural for PBO).

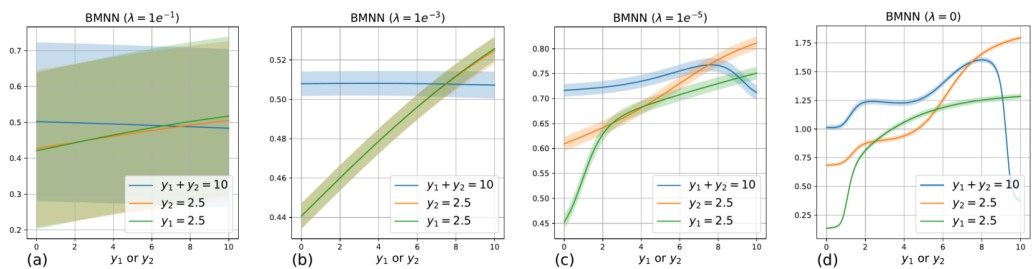

Figure 13: Bayesian Monotonic Neural Network training with different $\lambda$ given 15 randomly sampled pairs from the slice $y_1 + y_2 = 10$ from the complicated utility function: a trade-off between the Hinge loss and uncertainty. A value of $\lambda = 1 \times 10^{-5}$ achieves a good balance between the prior and uncertainty.

Tables 1 and Tables 2 present the model uncertainty and accuracy evaluations in the following experimental setup: For problems of dimension $d < 5$, we use 160 samples, while for $d \geq 5$, we use 320 samples. The initial number of pairwise comparisons is set to 10 for $d < 5$ and 20 for $d \geq 5$, with a utility noise level of 0.1. After constructing the utility surrogate model, we evaluate its performance by randomly selecting pairs and comparing the model's predictions against the true preferences. This process is repeated 50 times. Table 3 reports the training time for the three utility surrogate models under these experimental conditions, with all computations performed on an Intel Core i7-1355U CPU.

In addition, we visualize an example of the optimization trajectories of qNEHVI, BOPE-GP, and BOPE-MoNNE on the ZDT1 test problem over 25 iterations (steps 1-25). As shown in Figure 14(a), qNEHVI selects points that maximize hypervolume, but these chosen points can be distant from optimal solutions. According to Figure 14(b), BOPE-GP gradually selects points closer to the Pareto front but remains far from the optimal regions. While allowing BOPE-GP to compare arbitrary output pairs (rather than only observed ones) improves its performance (Figure 14(c)), it still underperforms compared to BOPE-MoNNE (Figure 14(d)), which directly converges toward optimal points. We further demonstrate in Appendix A.8 that even when BOPE-GP is allowed arbitrary output comparisons, it continues to perform worse than BOPE-MoNNE.

Table 1: Assessment of the quality of the uncertainty estimates provided by the models

|  | GP | MoNNE | BMNN (VI) |
|---|---|---|---|
| DTLZ2 | 0.622(±0.01) | 0.924(±0.014) | 0.855(±0.019) |
| VLMOP3 | 0.625(±0.01) | 0.915(±0.022) | 0.731(±0.027) |
| ZDT1 | 0.674(±0.012) | 0.951(±0.025) | 0.911(±0.034) |
| VehicleSafety | 0.63(±0.016) | 0.863(±0.045) | 0.848(±0.04) |
| OSY | 0.64(±0.015) | 0.878(±0.022) | 0.886(±0.021) |
| CarCabDesign | 0.658(±0.008) | 0.951(±0.08) | 0.911(±0.02) |

Table 2: Surrogate Model Accuracy (accuracy is measured as the percentage of correct predictions)

|  | GP | MoNNE | BMNN (VI) |
|---|---|---|---|
| DTLZ2 | 0.9 | 0.92 | 0.9 |
| VLMOP3 | 0.86 | 0.92 | 0.84 |
| ZDT1 | 0.92 | 0.96 | 0.94 |
| VehicleSafety | 0.82 | 0.86 | 0.86 |
| OSY | 0.82 | 0.88 | 0.88 |
| CarCabDesign | 0.84 | 0.96 | 0.94 |

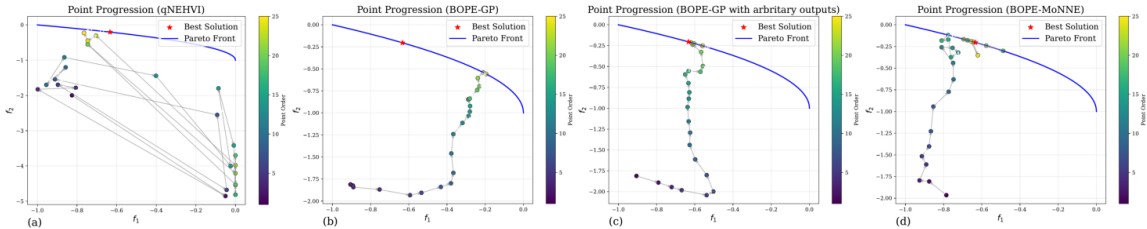

Figure 14: Optimization Trajectories of Different Algorithms

## A.6 Activation Function Comparison

The selection of an appropriate activation function is crucial to the performance of the BNNs [Li et al., 2024]. While prior work has compared smooth (Tanh) and non-smooth (ReLU) activation functions in BNNs, no clear consensus emerged regarding their relative effectiveness. In our investigation, we initially considered five activation functions in MoNNEs: Tanh, Sigmoid, Swish [Ramachandran et al., 2017], ReLU, and Leaky ReLU [Xu, 2015].

During the empirical investigation, we observed frequent numerical instabilities when using Tanh and ReLU activation functions, which led us to focus our analysis on three alternatives: Sigmoid, Swish, and Leaky ReLU. These functions provide an instructive comparison between smooth (Sigmoid, Swish) and non-smooth (Leaky ReLU) activation functions. For Leaky ReLU, the negative slope parameter is set to 0.25. Figure 16 presents the performance comparison of MoNNE in these activation

Table 3: Computation Time (seconds)

|  | GP | MoNNE | BMNN (VI) |
|---|---|---|---|
| DTLZ2 | 0.39(±0.02) | 3.53(±0.91) | 61.34(±0.17) |
| VLMOP3 | 0.42(±0.02) | 4.19(±0.91) | 60.66(±0.01) |
| ZDT1 | 0.40(±0.03) | 3.53(±1.09) | 63.52(±0.11) |
| VehicleSafety | 0.49(±0.04) | 5.58(±1.05) | 60.88(±0.02) |
| OSY | 0.49(±0.03) | 3.83(±0.8) | 123.59(±0.04) |
| CarCabDesign | 0.28(±0.01) | 4.92(±0.98) | 124.32(±0.11) |

functions. While all three activation functions demonstrate comparable performance, Swish exhibits marginally superior results.

Although Leaky ReLU achieves performance comparable to that of Swish in the previous experiments, it exhibits limitations in modelling complex utility functions. Figure 15 illustrates these limitations by displaying the base models of MoNNE with different activation functions, trained on 30 randomly sampled pairs from the constraint surface $y_1 + y_2 = 10$. We examine the same complex function previously analyzed in Figure 2(d) of Section 3, plotting the model values along the constraint surface $y_1 + y_2 = 10$. The visualization reveals that Leaky ReLU struggles to capture the intricate patterns of the underlying function.

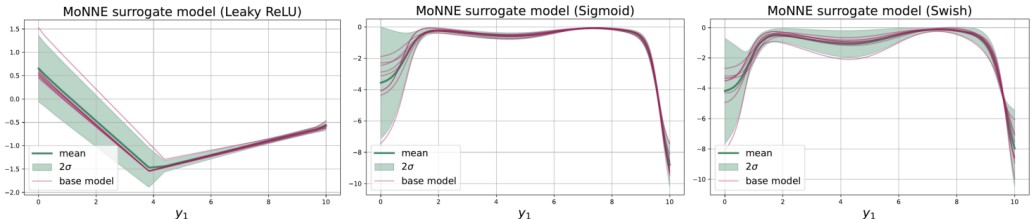

Figure 15: Given the shallow architecture of the neural network, Leaky ReLU activation functions may not be ideal for modelling complex underlying functions.

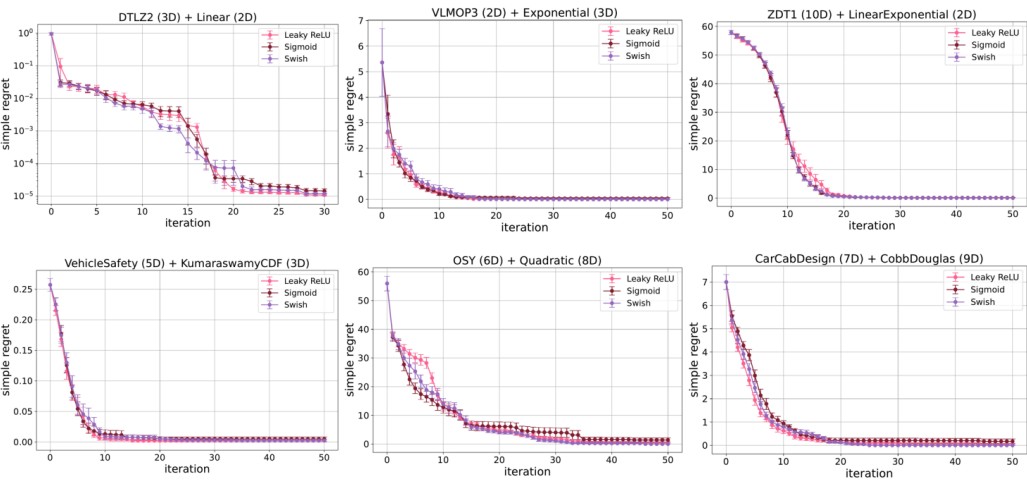

Figure 16: While all three activation functions demonstrate comparable performance, Swish exhibits marginally superior results.

## A.7 Layer Number Comparison

In this section, we analyze the relationship between network depth and model performance. We evaluate architectures with two layers ($d \times 100, 100 \times 1$), three layers ($d \times 100, 100 \times 10, 10 \times 1$), and four layers ($d \times 100, 100 \times 100, 100 \times 10, 10 \times 1$). Figure 17 illustrates the behavior of MoNNE models trained on 30 randomly sampled pairs from the constraint surface $y_1 + y_2 = 10$, using the complicated utility function described in Section 3. Figure 17 reveals that the learned representations maintain consistent patterns across varying network depths, with no clear correlation between the number of layers and the model's mean and variance.

We evaluate three different network depths across six test problems, as shown in Figure 18. While the optimal number of layers varies across problems, we observe that the 3-layer MoNNE consistently achieves intermediate performance. Notably, the parameter count increases substantially with network depth: the 2-layer MoNNE contains hundreds of parameters, the 3-layer variant introduces approximately 1,000 additional parameters, and the 4-layer architecture further adds 10,000 parameters. Moreover, the 4-layer MoNNE exhibits significantly longer training times compared to shallower

architectures. Based on this trade-off between performance and computational efficiency, we adopt the 3-layer MoNNE architecture for all subsequent experiments in this paper.

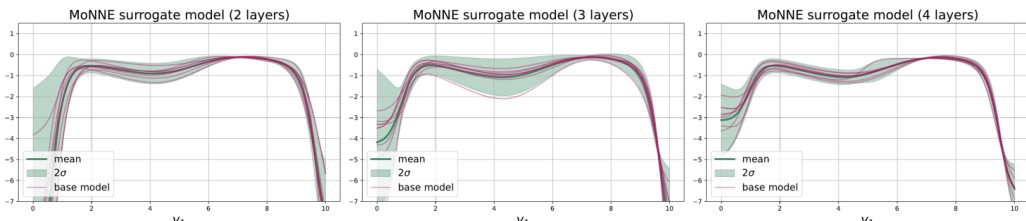

Figure 17: The patterns of MoNNEs with different layer numbers are similar.

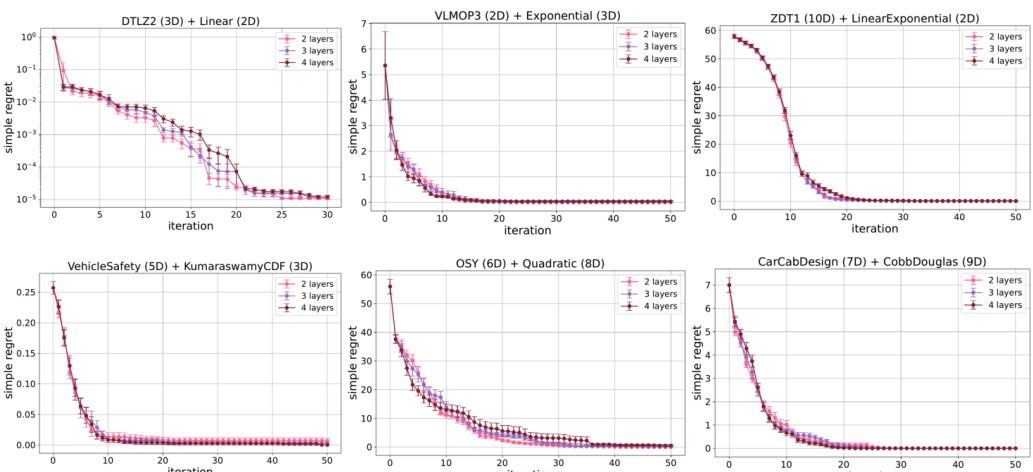

Figure 18: While the optimal number of layers remains unclear, the 3-layer MoNNE consistently achieves intermediate performance.

## A.8 EUBO Experiments

In this section, we compare two versions of the acquisition function EUBO, one that can choose arbitrary outputs for comparison, and one that is restricted to choose among previously observed outputs. Both use a GP as utility surrogate model. The number of posterior samples of the GP is set as $N = 32$. In addition, the utility noise is set to $0.1$.

In Figure 19, EUBO (arbitrary output) works better than EUBO (observed output) given the GP models. However, MoNNE+IEUBO still works the best.

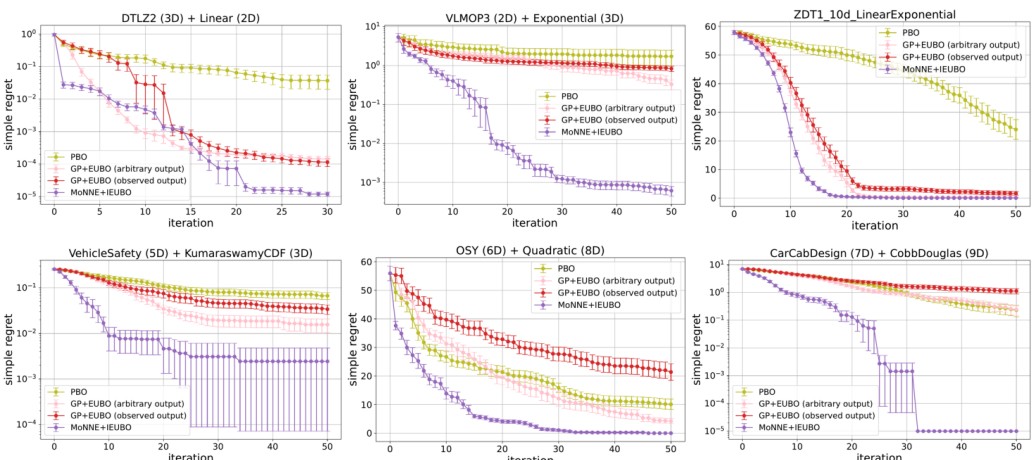

Figure 19: Allowing arbitrary outputs in EUBO for GPs improves the BOPE-GP performance, yet it still remains inferior to BOPE-MoNNE.

In Figure 20, we compare IEUBO and EUBO with and without utility noise and given the MoNNE model. Our experiments demonstrate that IEUBO generally outperforms EUBO in the presence of utility noise. However, when utility noise is absent, EUBO achieves comparable or occasionally superior performance to IEUBO.

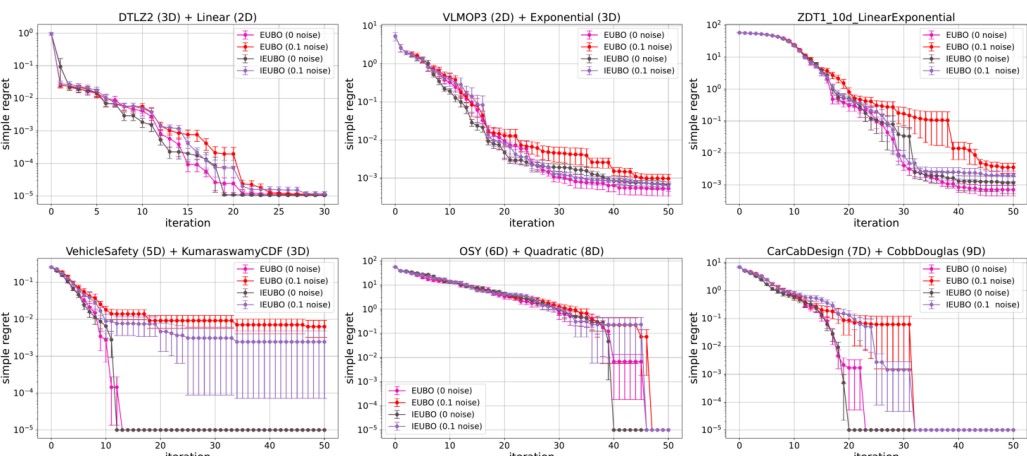

Figure 20: IEUBO in most cases outperforms EUBO in the presence of utility noise, but when utility noise is absent, EUBO achieves comparable or occasionally superior performance to IEUBO.

## A.9    Influence of Utility Noise

In Figure 21, we show the number of preference errors that the DM made during the experiment. Note that the error number for the VehicleSafety problem and the CarCabDesign problem plateaus due to several runs reaching the error threshold and stopping.

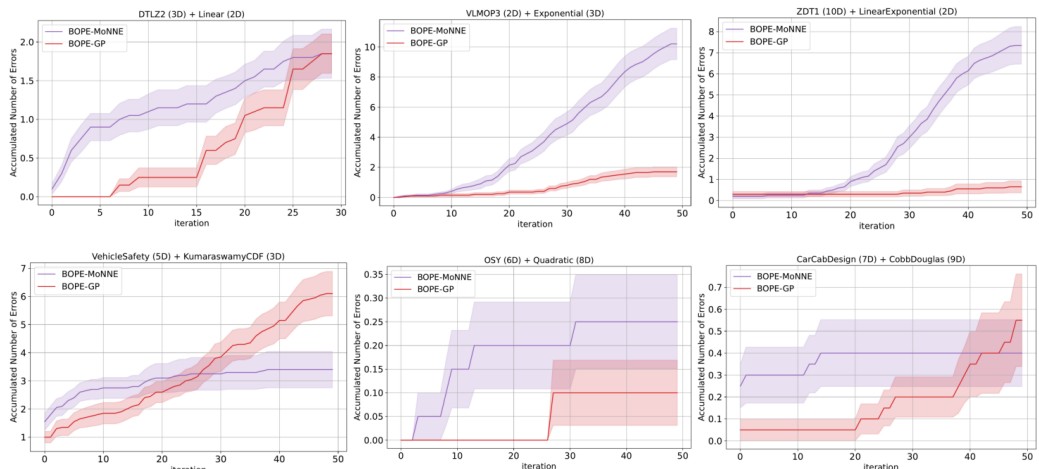

Figure 21: Comparison of Preference Errors in BOPE-GP and BOPE-MoNNE. BOPE-MoNNE generally has more errors. Notably, in the VehicleSafety problem and the CarCabDesign problem, the error count for BOPE-MoNNE plateaus due to several runs reaching the error threshold and stopping.

We conducted experiments with different noise levels ($\sigma_{\text{noise}} = 0$, $\sigma_{\text{noise}} = 0.1$ and $\sigma_{\text{noise}} = 0.5$) to investigate the impact of noise on both the GP model and the MoNNE model, with results shown in Figure 22. Our analysis reveals that BOPE-MoNNE consistently outperforms BOPE-GP across all noise levels. This performance difference is particularly noteworthy in the VehicleSafety problem, where a noise level of $\sigma_{\text{noise}} = 0.5$ represents significant disruption, given that the output range of this problem is approximately $0.6$. Under these conditions, pairwise comparisons become nearly random. While BOPE-GP performance deteriorates substantially in this high-noise scenario, BOPE-MoNNE maintains reasonable effectiveness by preserving the underlying monotonicity constraints of the problem.

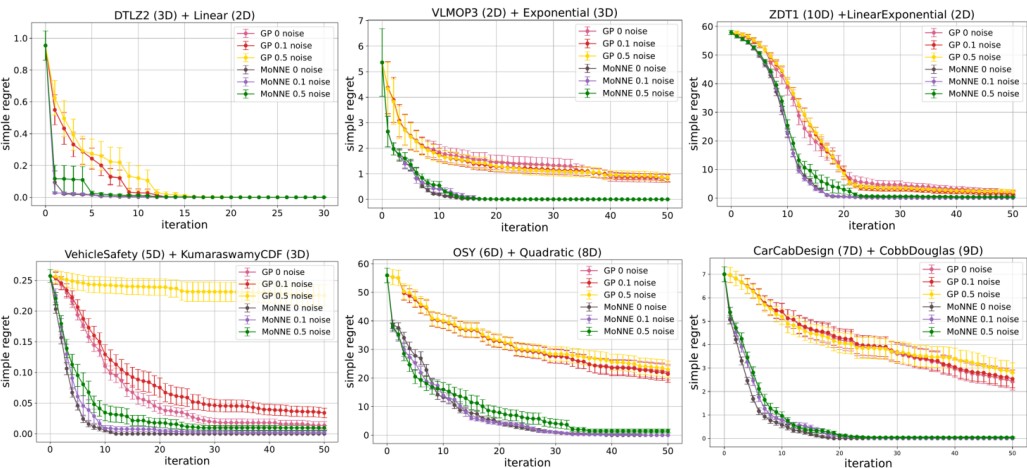

Figure 22: MoNNE models are robust under utility noise.

## A.10 Derivation of Expected Maximum of Two Normal Distributions

This result can also be found in Nadarajah and Kotz [2008].

Let $X \sim \mathcal{N}(\mu_1, \sigma_1^2)$ and $Y \sim \mathcal{N}(\mu_2, \sigma_2^2)$ be independent normal random variables. We derive $\mathbb{E}[\max(X, Y)]$ as follows:

First, we apply the law of total expectation:
$$\mathbb{E}[\max(X, Y)] = \mathbb{E}[X|X > Y]\mathbb{P}(X > Y) + \mathbb{E}[Y|Y > X]\mathbb{P}(Y > X).$$

Define $Z = X - Y$. Since $X$ and $Y$ are independent normals:
$$Z \sim \mathcal{N}(\mu_1 - \mu_2, \sigma_1^2 + \sigma_2^2).$$

Let $\sigma_3 = \sqrt{\sigma_1^2 + \sigma_2^2}$ and $\alpha = \frac{\mu_1 - \mu_2}{\sigma_3}$.

The probabilities are:
$$\mathbb{P}(X > Y) = \mathbb{P}(Z > 0) = \Phi(\alpha),$$
$$\mathbb{P}(Y > X) = \mathbb{P}(Z < 0) = \Phi(-\alpha) = 1 - \Phi(\alpha),$$

where $\Phi(\cdot)$ is the standard normal CDF.

We first need to calculate:
$$\mathbb{E}[X|X > Y] = \mathbb{E}[X|Z > 0],$$

where $\phi(\cdot)$ is the standard normal PDF.

$(X, Z)$ follows a bivariate normal:
$$\begin{pmatrix} X \\ Z \end{pmatrix} \sim \mathcal{N}\left( \begin{pmatrix} \mu_1 \\ \mu_1 - \mu_2 \end{pmatrix}, \begin{pmatrix} \sigma_1^2 & \sigma_1^2 \\ \sigma_1^2 & \sigma_1^2 + \sigma_2^2 \end{pmatrix} \right).$$

The correlation coefficient between $X$ and $Z$ is $\rho = \frac{\sigma_1^2}{\sigma_1\sqrt{\sigma_1^2 + \sigma_2^2}} = \frac{\sigma_1}{\sigma_3}$.

For bivariate normal, $\mathbb{E}[X|Z = z]$ is linear in $z$:
$$\mathbb{E}[X|Z = z] = \mu_1 + \rho\frac{\sigma_1}{\sigma_3}(z - \mu_z),$$

where $\mu_z = \mu_1 - \mu_2$.

Hence,
$$\begin{aligned} \mathbb{E}[X|Z > 0] &= \mathbb{E}[\mathbb{E}[X|Z]|Z > 0] \\ &= \mathbb{E}[\mu_1 + \rho\frac{\sigma_1}{\sigma}(Z - \mu_z)|Z > 0] \\ &= \mu_1 + \rho\frac{\sigma_1}{\sigma}(\mathbb{E}[Z|Z > 0] - \mu_z) \\ &= \mu_1 + \rho\frac{\sigma_1}{\sigma}(\mu_z + \sigma\frac{\phi(\alpha)}{\Phi(\alpha)} - \mu_z) \\ &= \mu_1 + \rho\sigma_1\frac{\phi(\alpha)}{\Phi(\alpha)} \\ &= \mu_1 + \frac{\sigma_1^2}{\sigma_3}\frac{\phi(\alpha)}{\Phi(\alpha)}. \end{aligned}$$

That is, $\mathbb{E}[X|X > Y] = \mu_1 + \frac{\sigma_1^2}{\sigma_3}\frac{\phi(\alpha)}{\Phi(\alpha)}$.

Similarly:
$$\mathbb{E}[Y|Y > X] = \mu_2 + \frac{\sigma_2^2}{\sigma_3}\frac{\phi(-\alpha)}{\Phi(-\alpha)}.$$

Substituting back:
$$\mathbb{E}[\max(X, Y)] = \left(\mu_1 + \frac{\sigma_1^2}{\sigma_3}\frac{\phi(\alpha)}{\Phi(\alpha)}\right)\Phi(\alpha) + \left(\mu_2 + \frac{\sigma_2^2}{\sigma}\frac{\phi(-\alpha)}{\Phi(-\alpha)}\right)\Phi(-\alpha).$$

The closed-form expression is:
$$\boxed{\mathbb{E}[\max(X, Y)] = \mu_1\Phi(\alpha) + \mu_2\Phi(-\alpha) + \sigma_3\phi(\alpha)}.$$

