# OpenReview forum: "Bayesian Optimization with Preference Exploration using a Monotonic Neural Network Ensemble"
_NeurIPS.cc/2025/Conference — NeurIPS 2025 poster_

### Official Review · Reviewer_v6yY · 2025-06-18

**Clarity:** 3
**Significance:** 3
**Originality:** 3
**Rating:** 5
**Confidence:** 3

**Summary:**

This paper introduces BOPE-MoNNE (Bayesian Optimization with Preference Exploration via Monotonic Neural Network Ensemble), a novel multi-objective optimization method that learns decision-makers’ preferences by leveraging the monotonicity of underlying utility functions. A key advantage of the proposed approach is its ability to incorporate utility monotonicity through neural network design—an often-overlooked yet critical aspect of preference modeling. Extensive experiments demonstrate the efficiency of the proposed method.

**Questions:**

1. Regarding lines 161–162: While constraining preference queries to only previously observed output pairs enhances realism, could this restriction hinder the efficiency of preference learning by limiting exploration?
2. The ablation study section is completely moved to the appendix and somewhat limit. I would encourage the authors to include it in the main text, because the ablation study provides important insights into the contribution of key components of the proposed method.
3. MoNNE requires training an ensemble of 8 networks, which is more expensive than GP-based methods. Is the performance gain worth the computational cost?

**Ethical Concerns:**

["NO or VERY MINOR ethics concerns only"]

**Final Justification:**

The review's detailed response helped clarify my concerns. I will keep my score unchanged.

**Limitations:**

yes

**Paper Formatting Concerns:**

This paper does not have major formatting issues.

**Quality:**

3

**Strengths And Weaknesses:**

Strengths:
1. The proposed method uses neural networks as the utility surrogate model and incorporates the monotonicity of utility functions.
2. This paper proposes one modification acquisition function IEUBO for MoNNEs, experiments demonstrate that IEUBO generally outperforms EUBO in the presence of utility noise.
3 . The experimental results are comprehensive and verify the effectiveness of the proposed method.

Weaknesses:
While the paper demonstrates strong performance on synthetic benchmarks, the evaluation does not include real-world applications. A discussion of how the method might adapt to such settings—or a case study on real-world data—would strengthen the practical relevance of the contributions.

---

> ### Author Rebuttal · Authors · 2025-07-30
>
> Thank you for your thoughtful review and insightful comments. We provide detailed responses to your questions below, and will make corresponding changes to the camera-ready version of the paper.
>
> >the evaluation does not include real-world applications
>
> We would like to highlight that VehicleSafety, OSY, and CarCabDesign are generally considered real-world applications. Problems with a real decision maker are not considered as they would not be reproducible.
>
>
> >While constraining preference queries to only previously observed output pairs enhances realism, could this restriction hinder the efficiency of preference learning by limiting exploration?
>
> Asking a real decision maker to compare solutions that don’t exist may lead to dissatisfaction, which is why we do not allow it. But theoretically, allowing the comparison of arbitrary outputs may improve efficiency slightly, as we have demonstrated for BOPE-GP in Figure 17. For BOPE-MoNNE, allowing arbitrary outputs was slightly worse. In any case, BOPE-MoNNE which is constrained to historical observations still achieves superior performance compared to BOPE-GP which is allowed to use arbitrary outputs.
>
> > include ablation study in the main text
>
> Thank you for the suggestion! We placed this content in the appendix due to space constraints. We will revise the paper layout to include at least a paragraph highlighting the main results in the main text.
>
> >Is the performance gain worth the computational cost of an ensemble of 8 networks?
>
> This depends on the cost of training the MoNNE vs. the cost of running a simulation or asking an expert to provide preference information. Additionally, although MoNNE requires training 8 networks, we are able to  train them in parallel rather than sequentially, which provides significant speedup.

---

> > ### Comment · Reviewer_v6yY · 2025-08-03
> > **Official Comment by Reviewer v6yY**
> >
> > Thank you for the authors' response, and I will keep my score unchanged.

---

### Official Review · Reviewer_cbyB · 2025-06-26

**Clarity:** 3
**Significance:** 2
**Originality:** 3
**Rating:** 3
**Confidence:** 4

**Summary:**

This paper studies the problem of mulit-objective Bayesian optimization with preference exploration. It proposes a Monte-Carlo based method to solve the problem, using surrogate model of monotonic neural network ensemble. Experimental results indicate that it can achieve superior performance as compared to other baseline methods.

**Questions:**

1. What is the benefit of doing separate modelling for g and f? What if we just run preferential BO over the input to preference composite function? What is the intuition why the proposed method outperform PBO in Fig. 3？
2. Can the method be extended to the popular Bradley-Terry preference model?
3. How is the NN initialized? I expect this initialization method determines the prior of the model. (Similar to the lengthscale of the GP model.)
4. In Sec. 4.1, "Firstly, neural networks could be interpreted as nonparametric when their number of parameters is large enough, providing the desired flexibility." I feel this claim is a bit too strong.
5. What do you specifically mean by "scale invariance in preference learning"?

**Ethical Concerns:**

["NO or VERY MINOR ethics concerns only"]

**Limitations:**

1. NN usually requires more data to train. While BO setting has quite small dataset (especially in the initial steps). Can the authors comment on this issue?
2. The survey would be more complete with more recent works on preferential BO, since similar techniques can be easily transferred to multi-objective setting with theoretical guarantees.
3. The justification for only using $y$ in the historical data set for comparison seems weak to me. Indeed, if the preference elicitation is cheap, we can even build a preference model for $g$ offline.
4. Some typos and font issues:
- Line 28, "it may" ---> "it may be"
- Fig. 3, the font for xticks and yticks look small.

**Paper Formatting Concerns:**

I did not find any.

**Quality:**

3

**Strengths And Weaknesses:**

Quality: Overall the proposed method is solid with convincing experimental result. But there are some key concerns I feel the paper needs to address.
Clarity: The paper is mostly well-written except some minor typo and font issues.
Significance: The paper combines several existing techniques. The siginificance of the contribution is medium to me.
Originality: As far as I know, this work is original.

---

> ### Author Rebuttal · Authors · 2025-07-30
>
> Thank you for your thoughtful review and insightful comments. We provide detailed responses to your questions below and include the additional experimental results according to your suggestion. We will modify the camera-ready paper accordingly if the paper is accepted.
>
>
> > there are some key concerns I feel the paper needs to address
>
> We are not sure which key concerns the reviewer is referring to, as only minor typo and font issues as well as medium significance are mentioned.
>
>
> >What is the benefit of doing separate modelling for g and f? What if we just run preferential BO over the input to preference composite function? What is the intuition why the proposed method outperform PBO in Fig. 3?
>
> Our results show that PBO alone is not as good. Compared to PBO, our algorithm exploits the inherent structure of the problem. For nested BO problems $\text{max}~g(h(x))$ where $h$ is expensive-to-evaluate and $g$ is cheap to compute, modeling $h$ separately with GPs should outperform modeling $g(h(x))$ as a single GP, according to [1].
>
> Besides the nested structure, our approach also allows to leverage the monotonicity of the utility function, which in turn allows to recognize dominance in multi-objective optimization. The benefit of explicit monotonicity is also clearly visible in our ablation study.
> Therefore, BOPE-MONNE's superiority over PBO for this type of problem can be expected.
>
> [1] Astudillo, R., & Frazier, P. (2019, May). Bayesian optimization of composite functions. In International Conference on Machine Learning (pp. 354-363). PMLR.
>
>
> >Can the method be extended to the popular Bradley-Terry preference model?
>
> Thank you for this comment. The Bradley-Terry model is widely used to model choice behaviour in pairwise comparisons. It is, however, less a utility model (which aggregates different objective values into a scalar utility value), than a choice model which specifies probabilities of selecting one or the other alternative given their underlying utilities. We model this in our paper by adding a normally distributed random value to the true utility difference. Nevertheless, the two show very similar behavior in pairwise comparisons.
>
> The Bradley-Terry preference model defines the probability of preferring item $i$ over item $j$, given their utility values $U_i$ and $U_j$, as $P(i \succ j) = \frac{e^{U_i}}{e^{U_i} + e^{U_j}} = \frac{1}{1 + e^{-\delta}}$, where $\delta = U_i - U_j$ denotes the utility gap. A fundamental limitation of the standard Bradley-Terry model is its sensitivity to utility scale. For example, if a utility function ranges over $[0, 1]$ and we scale all utility values by 10 to obtain a new function ranging over $[0, 10]$, the predicted probabilities change significantly, even though the preference ordering remains the same. To address this issue, a more flexible formulation introduces a scale parameter $\beta$, yielding $P(i \succ j) = \frac{1}{1 + e^{-\beta \delta}}$.
> With the additive Gaussian noise used in our experiments, i.e., $\epsilon \sim \mathcal{N}(0, \sigma^2)$, the probability of selecting item $i$ becomes $P(U_i - U_j + \epsilon > 0) = \Phi\left(\frac{\delta}{\sigma}\right)$, where $\Phi(\cdot)$ denotes the standard normal cumulative distribution function.
>
> Both the Bradley-Terry and Gaussian choice models exhibit similar behavior: as $\delta \to 0$, the selection probability approaches $0.5$, and as $\delta \to \infty$, it approaches $1$. Moreover, for a given noise level $\sigma$, one can choose a corresponding $\beta$ such that the two models closely align. For instance, when $\sigma = 1$, setting $\beta = 1.8$ results in almost identical probability curves. Therefore, we expect that our results will not change if our Gaussian-noise based formulation would be replaced by the very similar Bradley-Terry choice model. We will mention the similarity of the two models in our paper, as indeed the Bradley-Terry model is very popular and making this link is helpful for many readers.
>
> >How is the NN initialized?
>
> The initial parameters are set by:
>
>         stdv = 1. /self.weight.size(1)
>         self.weight.data.uniform_(-stdv-6, stdv)
>         self.bias.data.uniform_(-stdv, stdv)
>
> The bias initialization follows standard practice. For weights, we use a smaller range with a shifted lower bound (-stdv-6 instead of -stdv) because the subsequent exponential transformation amplifies the values. We will mention it in the revised paper.
>
> >"Firstly, neural networks could be interpreted as nonparametric when their number of parameters is large enough, providing the desired flexibility." I feel this claim is a bit too strong.
>
> ANNs are universal function approximators [1]. But we agree with the reviewer that our claim was a bit too strong. We have revised the sentence to: "First, neural networks with sufficient parameters can approximate complex utility functions without requiring explicit assumptions about their functional form, providing the desired modeling flexibility."
>
> [1] Hornik, K., Stinchcombe, M. and White, H., 1989. Multilayer feedforward networks are universal approximators. Neural networks, 2(5), pp.359-366.
>
>
> >meaning of "scale invariance in preference learning"?
>
> Just from pairwise preference information, we cannot uniquely determine the underlying utility function. Any monotonic transformation will result in the same preference statements.
>
>
> >NN usually requires more data to train. While BO setting has quite small dataset (especially in the initial steps).
>
> As our paper shows in Appendix A.3, given pairwise comparisons, if the network is trained using Hamiltonian Monte Carlo, the size of the dataset is indeed a problem. For this reason, we use the ensemble method.
>
> Neural networks have been used as surrogate models also in other traditional BO settings, such as [1][2][3].  The key difference of NNs in BO (when compared with NNs in Natural Language Processing or Computer Vision) is their typical shallow architecture (3-4 layers), which reduces data requirements and makes them well-suited for BO's limited data regime.
>
> Besides, we do compare to strong baselines designed for a low data regime (such as GP-based methods), and our approach works better.
>
> [1] Snoek, J., Rippel, O., Swersky, K., Kiros, R., Satish, N., Sundaram, N., ... & Adams, R. (2015, June). Scalable bayesian optimization using deep neural networks. In International Conference on Machine Learning (pp. 2171-2180). PMLR.
>
> [2] Li, Y. L., Rudner, T. G. J., & Wilson, A. G. (2024). A study of Bayesian neural network surrogates for Bayesian optimization. In The Twelfth International Conference on Learning Representations. https://openreview.net/forum?id=SA19ijj44B
>
> [3] Brunzema, P., Jordahn, M., Willes, J., Trimpe, S., Snoek, J., & Harrison, J. (2025). Bayesian optimization via continual variational last layer training. In The Thirteenth International Conference on Learning Representations. https://openreview.net/forum?id=1jcnvghayD
>
>
> >more recent works on preferential BO
>
> We now cover a few more references in the related work section:
>
> Xu, W., Wang, W., Jiang, Y., Svetozarevic, B. and Jones, C.N., 2024. Principled preferential Bayesian optimization. arXiv preprint arXiv:2402.05367.
>
> Arun Kumar, A. V., Shilton, A., Gupta, S., Rana, S., Greenhill, S., & Venkatesh, S. (2024, August). Enhanced Bayesian optimization via preferential modeling of abstract properties. In Joint European Conference on Machine Learning and Knowledge Discovery in Databases (pp. 234-250). Cham: Springer Nature Switzerland.
>
> Mikkola, P., Todorović, M., Järvi, J., Rinke, P., & Kaski, S. (2020, November). Projective preferential bayesian optimization. In International Conference on Machine Learning (pp. 6884-6892). PMLR.
>
> However, preferential BO doesn’t benefit from information about the nested problem structure and monotonicity of the utility function and thus is expected to perform worse in our setting.
>
> >The justification for only using y in the historical data set for comparison seems weak. If the preference elicitation is cheap, we can even build a preference model for g offline.
>
> We can extend pairwise comparison to arbitrary outputs. In our paper we have decided to only compare outputs from historical observations because we are concerned that presenting non-existent options to a decision maker might lead to complaints in practice, when a solution examined and desired by the decision maker doesn’t exist.
>
> Learning preferences is usually not cheap, as it requires time from a human expert. Besides, without knowledge of the output values or ranges, offline preference learning becomes challenging as we cannot determine which outputs to present for comparison.
>
> >Some typos and font issues
>
> Thank you for carefully pointing this out. We have corrected the typo and updated Figure 3 by adjusting the font size of the x-axis and y-axis tick labels in the revised version.

---

### Official Review · Reviewer_jmqt · 2025-07-01

**Clarity:** 3
**Significance:** 2
**Originality:** 3
**Rating:** 4
**Confidence:** 4

**Summary:**

The paper introduces BOPE-MoNNE, a method for Bayesian Optimization with Preference Exploration (BOPE) that learns a decision maker’s preferences from pairwise comparisons. BOPE-MoNNE models the utility function with a monotonic neural network ensemble, which enforces monotonicity through exponential weight transformations and trains on pairwise data. They show neural network ensembles provide better-calibrated uncertainty estimates than Bayesian neural networks, which struggle with scale invariance in pairwise preference learning. The method adapts standard acquisition functions. Across five benchmark problems, BOPE-MoNNE consistently outperforms previous methods, demonstrating that incorporating monotonicity and ensemble-based uncertainty estimation significantly improves performance.

**Questions:**

How does the method perform over a larger variety of utility functions and benchmarks?

Are there cases where BOPE-MoNNE fails/performs worse than baselines?

**Ethical Concerns:**

["NO or VERY MINOR ethics concerns only"]

**Final Justification:**

The authors have cleared my concerns. Keeping the original score.

**Limitations:**

The authors could further address the limitations of their work. They mention the additional computational overhead but do not discuss any other possible limitations.

**Quality:**

2

**Strengths And Weaknesses:**

The method shows strong performance compared to baselines on the test functions presented. Additionally, their contributions include adding monotonicity to the utility model, which could better mimic real-world scenarios than other methods that do not enforce this condition. There are extension ablations provided in the appendix detailed network choices and comparisons to other ways of modeling utility.

However, the overall evaluation is fairly limited and more robust evaluation on a larger variety of benchmarks would be more convincing. This includes scaling to more dimensions or more complex utility functions, as well as a greater quantity of evaluations.

---

> ### Author Rebuttal · Authors · 2025-07-30
>
> Thank you for your thoughtful review and insightful comments. We provide detailed responses to your questions below and include the additional experimental results according to your suggestion. We will modify the camera-ready paper accordingly if the paper is accepted.
>
> > How does the method perform over a larger variety of utility functions and benchmarks?
>
> We evaluated our approach on five benchmark problems with dimensions ranging from 2 to 7 and number of objectives ranging from 2 to 9. This experimental setup is consistent with the scale and scope commonly used in the existing literature. In particular, we use all four multi-objective problems that were used in the benchmark paper [1], plus an additional one. Regarding the utility functions used, we already test a wide range of utility functions from [1] and [2], plus the Cobb-Douglas function which is a standard function in economics. We have not observed a significant influence of the utility function on results, and wouldn’t expect so since ANNs are at least in principle capable of representing any utility function.
>
> To enhance the comprehensiveness of our evaluation, we have added an additional test case: the well-known multi-objective benchmark ZDT1 with 10 dimensions and 2 objectives, using a linear-exponential utility function defined as $U(y_0,y_1) = 5(y_0+2) + 5(y_1+2) + 2\exp(0.75(y_0+2))\exp(1.25(y_1+2))$, i.e., a sum of a linear component and an exponential product. The experimental result (20 repetitions, 50 iterations and all the experiment setting follows the paper) is shown in the table below:
>
> | Method        | 25th                 | 50th                 | 75th                 | 100th                |
> |---------------|----------------------|----------------------|----------------------|----------------------|
> | Random        | 57.58 (0.6948)       | 57.01 (0.6948)       | 56.75 (0.5361)       | 56.75 (0.5361)       |
> | qNEHVI        | 35.15 (1.514)        | 11.77 (1.913)        | 6.319 (1.136)        | 4.816 (1.047)        |
> | qParEGO       | 51.72 (0.7307)       | 48.53 (1.331)        | 47.80 (1.322)        | 46.44 (1.273)        |
> | PBO           | 52.62 (1.496)        | 48.01 (1.886)        | 38.66 (2.450)        | 24.75 (3.359)        |
> | BOPE-Linear   | 13.78 (1.283)        | 0.8842 (0.1596)      | 0.8842 (0.1596)      | 0.8842 (0.1596)      |
> | BOPE-GP       | 32.43 (2.314)        | 3.403 (0.8208)       | 2.327 (0.6852)       | 1.674 (0.6571)       |
> | BOPE-BMNN     | 14.08 (1.799)        | 0.3960 (0.0804)      | 0.2231 (0.0439)      | 0.2231 (0.0439)      |
> | BOPE-MoNNE    | 9.588 (0.8750)       | 0.2127 (0.0899)      | 0.0025 (0.0010)      | 0.0020 (0.0007)      |
> | Utility Known | 8.676 (1.016)        | 0.0558 (0.0183)      | 0.000125 (0.000034)  | 0.000059 (0.000018)  |
>
>
> From the table, we observe that BOPE-MoNNE achieves the best performance, followed by BOPE-BMNN and BOPE-Linear. The strong performance of the linear model can be attributed to the structure of the utility function, where the first component is linear. While BOPE-GP performs worse than these three methods, it still significantly outperforms the remaining baselines. PBO exhibits weaker performance, which can be explained by the challenge of navigating a 10d space using only pairwise comparisons.
>
> We also conducted an ablation study as shown below (same experimental setting):
>
> | Method                          | 25th                   | 50th                   | 75th                   | 100th                  |
> |----------------------------------|-------------------------|-------------------------|-------------------------|-------------------------|
> | BOPE-GP                          | 32.43 (2.314)           | 3.403 (0.8208)          | 2.327 (0.6852)          | 1.674 (0.6571)          |
> | BOPE-MoNNE (monotonicity removed) | 17.98 (1.787)           | 1.275 (0.3492)          | 0.2872 (0.1461)         | 0.09786 (0.08952)       |
> | BOPE-MoNNE (ensemble removed)     | 9.195 (1.257)           | 0.4128 (0.1342)         | 0.2087 (0.1192)         | 0.1204 (0.0893)         |
> | BOPE-MoNNE                        | 9.588 (0.8749)          | 0.2127 (0.0899)         | 0.0025 (0.0010)         | 0.0020 (0.0007)         |
>
>
> The results demonstrate that  incorporating monotonicity information improves the performance significantly, achieving better function values at a lower cost.
>
>
>
> [1] Lin, Z.J., Astudillo, R., Frazier, P. and Bakshy, E., 2022, May. Preference exploration for efficient Bayesian optimization with multiple outcomes. In International Conference on Artificial Intelligence and Statistics (pp. 4235-4258). PMLR.
> [2] Astudillo, R., & Frazier, P. (2020, June). Multi-attribute Bayesian optimization with interactive preference learning. In International Conference on Artificial Intelligence and Statistics (pp. 4496-4507). PMLR
>
> >Are there cases where BOPE-MoNNE fails/performs worse than baselines?
>
> We have not seen such examples and can’t think of a particular case where BOPE-MoNNE would perform worse. Conceptually and as shown in an ablation study, the incorporation of monotonicity should be beneficial, and the MoNNE seems to work well as a surrogate model. An ablation study also shows that the performance is robust with respect to small algorithmic variations.
>
>
>
>
> >discuss any other possible limitations
>
> We cannot think of any major limitations relative to the benchmark algorithms. MoNNE is a black box, so one doesn’t really know what utility function the algorithm has learned - but so is the GP used in the benchmark algorithm.
> Our study doesn’t involve any human decision makers, but doing so would not only be prohibitively expensive but also not be reproducible. Instead, we use artificial decision makers that share characteristics with human decision makers, in particular sometimes selecting the less preferred solution if the solutions have similar utility.
> The algorithm assumes a single stakeholder, and an extension to multiple stakeholders would be interesting - but other methods assume a single stakeholder as well. We will mention multiple stakeholders in the future work section.

---

> > ### Comment · Reviewer_jmqt · 2025-08-08
> > **Thank you for your response**
> >
> > Thank you for providing additional results and thoughtful responses to my questions, which would be great to include in the revised paper. I have no other concerns at this moment.

---

> > > ### Author Response · Authors · 2025-08-08
> > >
> > > Thank you again for your valuable comment. We have updated the revised paper to include the new results.

---

### Official Review · Reviewer_3gJh · 2025-07-03

**Clarity:** 4
**Significance:** 3
**Originality:** 2
**Rating:** 5
**Confidence:** 3

**Summary:**

In interactive multi-objective optimization, the utility functions are usually monotonic. The paper considers this fact and introduces a new method BOPE-MoNNE. BOPE-MoNNE uses neural network ensemble model with positive weight constraints to ensure monotonicity while modeling the mean and variance. The model is trained on pairwise preferences queried from the decision maker (DM). Experimental results on several established low-dimensional benchmark problems demonstrate that BOPE-MoNNE outperforms all baselines, and achieves performance close to the ideal case where the true utility is known.

**Questions:**

Could you provide visualizations the Pareto front of different methods? This could offer additional insight into the performance and trade-offs.

**Ethical Concerns:**

["NO or VERY MINOR ethics concerns only"]

**Final Justification:**

Thank you for the response addressing each of the concerns. I have changed the score accordingly.

**Limitations:**

The main limitations are the potential inefficiency due to frequent DM queries and the unclear scalability of the method to higher-dimensional or more challenging scenarios.

**Paper Formatting Concerns:**

None.

**Quality:**

3

**Strengths And Weaknesses:**

Strengths

- The proposed method effectively incorporates the monotonicity property of utility functions, which is often overlooked in previous work.
- Experimental results show that the proposed approach consistently achieves lower simple regret compared to existing methods.
- The paper is clearly written and easy to follow.

Weaknesses

- The method requires a preference query from the DM at every iteration, which could become impractical or inefficient for complex or real-world scenarios where querying is expensive.
- Experiments are limited to a few standard low-dimensional tasks. It is unknown whether the method performs well on more challenging tasks with higher dimensions.
- Although the authors state that code is available in an anonymous GitHub repository, I was unable to find the link. The lack of accessible code may hinder reproducibility of the results.

---

> ### Author Rebuttal · Authors · 2025-07-30
>
> Thank you for your thoughtful review and insightful comments. We provide detailed responses to your questions below and include the additional experimental results according to your suggestion. We will modify the camera-ready paper accordingly if the paper is accepted.
>
> > The method requires a preference query from the DM at every iteration
>
> While we alternate between preference elicitation and optimization steps, it is straightforward to change the ratio, e.g. have one preference query every two optimization steps. In our paper, we already test this case (Figure 10) and we find that BOPE-MoNNE still works best.
>
> > whether the method performs well on more challenging tasks with higher dimensions
>
> We evaluated our approach on five benchmark problems with dimensions ranging from 2 to 7 and number of objectives ranging from 2 to 9. This experimental setup is consistent with the scale and scope commonly used in the existing literature. In particular, we use all four multi-objective problems that were used in the benchmark paper [1], plus an additional one. Regarding the utility functions used, we already test a wide range of utility functions from [1] and [2], plus the Cobb-Douglas function which is a standard function in economics. We have not observed a significant influence of the utility function on results, and wouldn’t expect so since ANNs are at least in principle capable of representing any utility function.
>
> To enhance the comprehensiveness of our evaluation, we have added an additional test case: the well-known multi-objective benchmark ZDT1 with 10 dimensions and 2 objectives, using a linear-exponential utility function defined as $U(y_0,y_1) = 5(y_0+2) + 5(y_1+2) + 2\exp(0.75(y_0+2))\exp(1.25(y_1+2))$, i.e., a sum of a linear component and an exponential product. The experimental result (20 repetitions, 50 iterations and all the experiment setting follows the paper) is shown in the table below:
>
> | Method        | 25th                 | 50th                 | 75th                 | 100th                |
> |---------------|----------------------|----------------------|----------------------|----------------------|
> | Random        | 57.58 (0.6948)       | 57.01 (0.6948)       | 56.75 (0.5361)       | 56.75 (0.5361)       |
> | qNEHVI        | 35.15 (1.514)        | 11.77 (1.913)        | 6.319 (1.136)        | 4.816 (1.047)        |
> | qParEGO       | 51.72 (0.7307)       | 48.53 (1.331)        | 47.80 (1.322)        | 46.44 (1.273)        |
> | PBO           | 52.62 (1.496)        | 48.01 (1.886)        | 38.66 (2.450)        | 24.75 (3.359)        |
> | BOPE-Linear   | 13.78 (1.283)        | 0.8842 (0.1596)      | 0.8842 (0.1596)      | 0.8842 (0.1596)      |
> | BOPE-GP       | 32.43 (2.314)        | 3.403 (0.8208)       | 2.327 (0.6852)       | 1.674 (0.6571)       |
> | BOPE-BMNN     | 14.08 (1.799)        | 0.3960 (0.0804)      | 0.2231 (0.0439)      | 0.2231 (0.0439)      |
> | BOPE-MoNNE    | 9.588 (0.8750)       | 0.2127 (0.0899)      | 0.0025 (0.0010)      | 0.0020 (0.0007)      |
> | Utility Known | 8.676 (1.016)        | 0.0558 (0.0183)      | 0.000125 (0.000034)  | 0.000059 (0.000018)  |
>
>
> From the table, we observe that BOPE-MoNNE achieves the best performance, followed by BOPE-BMNN and BOPE-Linear. The strong performance of the linear model can be attributed to the structure of the utility function, where the first component is linear. While BOPE-GP performs worse than these three methods, it still significantly outperforms the remaining baselines. PBO exhibits weaker performance, which can be explained by the challenge of navigating a 10d space using only pairwise comparisons.
>
> We also conducted an ablation study as shown below (same experimental setting):
>
> | Method                          | 25th                   | 50th                   | 75th                   | 100th                  |
> |----------------------------------|-------------------------|-------------------------|-------------------------|-------------------------|
> | BOPE-GP                          | 32.43 (2.314)           | 3.403 (0.8208)          | 2.327 (0.6852)          | 1.674 (0.6571)          |
> | BOPE-MoNNE (monotonicity removed) | 17.98 (1.787)           | 1.275 (0.3492)          | 0.2872 (0.1461)         | 0.09786 (0.08952)       |
> | BOPE-MoNNE (ensemble removed)     | 9.195 (1.257)           | 0.4128 (0.1342)         | 0.2087 (0.1192)         | 0.1204 (0.0893)         |
> | BOPE-MoNNE                        | 9.588 (0.8749)          | 0.2127 (0.0899)         | 0.0025 (0.0010)         | 0.0020 (0.0007)         |
>
>
>
> The results demonstrate that  incorporating monotonicity information improves the performance significantly, achieving better function values at a lower cost.
>
>
>
> [1] Lin, Z.J., Astudillo, R., Frazier, P. and Bakshy, E., 2022, May. Preference exploration for efficient Bayesian optimization with multiple outcomes. In International Conference on Artificial Intelligence and Statistics (pp. 4235-4258). PMLR.
> [2] Astudillo, R., & Frazier, P. (2020, June). Multi-attribute Bayesian optimization with interactive preference learning. In International Conference on Artificial Intelligence and Statistics (pp. 4496-4507). PMLR
>
> >  unable to find the code link
>
> The link is provided on line 71 of our paper and can be accessed by clicking "the anonymous GitHub repository". If the link is not functional, perhaps it was removed by the system to ensure anonymity, even though it was an anonymous GitHub repository. NeurIPS rules don’t allow us to provide a link in the rebuttal.
>
> > visualizations the Pareto front of different methods
>
> Thank you for your suggestion. NeurIPS rules don’t allow us to put figures into the rebuttal, but we already prepared a revised version of the paper with added visualization. We observed that MOBO methods (ParEGO and EHVI) exhibit more uniform exploration across the objective space, while BOPE methods concentrate their evaluations around high-utility regions. Furthermore, BOPE-MoNNE demonstrates closer convergence to the global optimum compared to BOPE-GP, as MoNNE provides better modeling of the utility function.
>
> > the potential inefficiency due to frequent DM queries and the unclear scalability of the method to higher-dimensional or more challenging scenarios
>
> As explained above, the paper already demonstrates that BOPE-MoNNE maintains its superiority if the DM is queried less frequently, and also on an additional 10-dimensional problem.

---

> > ### Comment · Reviewer_3gJh · 2025-08-07
> >
> > Thank you for the response addressing each of the concerns. I have changed the rating accordingly.

---

### Note · Authors · 2025-08-13

We thank all the reviewers for their valuable feedback and constructive suggestions. We have carefully addressed each of the comments and accordingly updated our paper.

Regarding the reviewer who gave an initial score of 3 and has not responded, we believe our rebuttal adequately addresses their concerns. Their main points relate to the use of DNNs in BO and the weaker performance of PBO compared to BOPE-MoNNE. As we explained, DNNs are well-established in BO and data sparsity is not an issue. BOPE-MoNNE, by design, leverages nested structures and monotonicity, giving it a clear advantage over PBO.

---

### Decision · Program_Chairs · 2025-09-17

**Decision:**

Accept (poster)

**Comment:**

This paper addresses multi-objective black-box optimization by integrating monotonicity into a neural network ensemble for preference learning. The main strengths lie in the clear presentation, comprehensive experiments, and demonstrated performance gains from exploiting monotonicity. The weaknesses concern limited evaluation on low-dimensional synthetic tasks and the potential impracticality of repeated preference queries in real-world scenarios. Nevertheless, all reviewers provided positive assessments, and the rebuttal effectively addressed most concerns. Overall, the contribution is solid and original, and I recommend acceptance, with the expectation that the authors incorporate reviewer feedback into the final version for improved completeness and clarity.